# Towards Win–Win Policies for Healthy and Sustainable Diets in Switzerland

**DOI:** 10.3390/nu12092745

**Published:** 2020-09-09

**Authors:** Alexi Ernstoff, Katerina S. Stylianou, Marlyne Sahakian, Laurence Godin, Arnaud Dauriat, Sebastien Humbert, Suren Erkman, Olivier Jolliet

**Affiliations:** 1Quantis SàRL, EPFL Innovation Park, Bat D, 1015 Lausanne, Switzerland; arnaud.dauriat@quantis-intl.com (A.D.); sebastien.humbert@quantis-intl.com (S.H.); ojolliet@umich.edu (O.J.); 2Department of Environmental Health Sciences, University of Michigan School of Public Health, 1415 Washington Heights, Ann Arbor, MI 48109, USA; kstylian@umich.edu; 3Institute for Sociological Research, University of Geneva, Boulevard Pont d’Arve 40, 1204 Geneva, Switzerland; marlyne.sahakian@unige.ch (M.S.); laurence.godin@fsaa.ulaval.ca (L.G.); 4Institute of Earth Surface Dynamics, University of Lausanne, Geopolis Building, 1015 Lausanne, Switzerland; suren.erkman@unil.ch

**Keywords:** disease burden, diet survey, vegetarian, vegan, sustainability, climate, gender

## Abstract

The first Swiss national dietary survey (MenuCH) was used to screen disease burdens and greenhouse gas emissions (GHG) of Swiss diets (vegan, vegetarian, gluten-free, slimming), with a focus on gender and education level. The Health Nutritional Index (HENI), a novel disease burden-based nutritional index built on the Global Burden of Disease studies, was used to indicate healthiness using comparable, relative disease burden scores. Low whole grain consumption and high processed meat consumption are priority risk factors. Non-processed red meat and dairy make a nearly negligible contribution to disease burden scores, yet are key drivers of diet-related GHGs. Swiss diets, including vegetarian, ranged between 1.1–2.6 tons of CO_2_e/person/year, above the Swiss federal recommendation 0.6 ton CO_2_e/person/year for all consumption categories. This suggests that only changing food consumption practices will not suffice towards achieving carbon reduction targets: Systemic changes to food provisioning processes are also necessary. Finally, men with higher education had the highest dietary GHG emissions per gram of food, and the highest disease burden scores. Win–win policies to improve health and sustainability of Swiss diets would increase whole grain consumption for all, and decrease alcohol and processed meat consumption especially for men of higher education levels.

## 1. Introduction

Food systems (activities ranging from agricultural production to consumption and waste) are major drivers of impacts on human health and the environment. In response, there is a growing need for food system transformation, with shifting diets as a key lever of transformational change. Specifically, in developed countries like Switzerland, shifting consumption away from animal-based foods (i.e., foods derived from animals) and towards more plant-based foods (i.e., foods derived from plants) is one of the main recommendations to meet sustainability goals from both health and environmental perspectives [1,2]. In addition, substantial amounts of literature focus on limiting consumption of highly processed foods and beverages which can lead to diets high in sodium and sugar [3,4,5]. Generally, consumers as well as policy makers are faced with a variety of messaging about healthy and sustainable diets from both scientific and lay sources [6].

There remains a major knowledge gap regarding how scientific information on health and sustainability influence perception and actual practices of consumers [7]. When it comes to advising policy change with science-based information there are also knowledge gaps. For example, there is a lack of simple screening methods (i.e., not complex epidemiological or health economics analyses) to support prioritization of policies, and to answer questions such as: for this population, is reducing salt or sugar more important? There is also a lack of reliable data on various dietary practices within a given population. Given limited data on actual practices, research on healthy and sustainable diets thereby tend to focus on (1) “average diets” extrapolated from national dietary surveys or market data on food (e.g., statistics from the Food and Agricultural Organization of the United Nations), (2) recommended diets by public officials (e.g., the Swiss food pyramid), and (3) hypothetical or optimized versions of diets e.g., assumed vegan or vegetarian diets [1,2,8,9].

The aim of this study is to assess the actual practices of diets that are perceived to be healthy and sustainable, rather than assuming hypothetical substitutions (e.g., for vegetarians) or recommended diets (e.g., food pyramid). To do this, we consider recent sociological research uncovering the pervasive perceptions of healthy and sustainable eating in Switzerland [6] and the actual practices reported in the first Swiss national dietary survey (Source: “Office fédéral de la sécurité alimentaire et des affaires vétérinaires: enquête nationale sur l’alimentation menuCH 2014/2015”) (MenuCH, 2017 [10]). We use metrics that allow the quantification and the relative comparison of health and environmental impacts between different diets. Life cycle assessment (LCA) was used to assess the greenhouse gas emissions (GHG) related to food production; to assess healthiness, we used the new Health Nutritional Index (HENI) that enables the disease burden characterization of dietary patterns by utilizing epidemiological evidence from the Global Burden of Disease study series [11,12]. Because this study focuses on different diets documented by MenuCH (which does not report the location of food production but only food product type), GHG was chosen as a robust and comparable sustainability indicator because it is less sensitive to location of food production unlike biodiversity and water impacts. Furthermore, GHG is relevant for current Swiss policy decisions regarding GHG targets. This is the first study to our knowledge that screens both the relative GHG and disease burden of the first Swiss survey data across various diets.

Through this work we aim to address the following questions: (a) What are the comparative magnitudes of disease burdens and GHG related to the average Swiss diet and other self-reported diets (e.g., vegetarian) in Switzerland? and (b) What factors (i.e., high or low consumption of certain foods or nutrients) are the main influencers of these impacts for various dietary patterns? Through answering these questions, we discuss the evidence for targeted policies to reduce diet-related disease burden and decrease GHG through dietary shifts in Switzerland.

## 2. Materials and Methods

### 2.1. Bridging the Gap between Perception and Practice

A first methodological step was to bridge the gap between the perceptions of healthy and sustainable diets in Switzerland versus what is actually practiced. To do so, we identified dominant “food prescriptions”, i.e., any guideline or recommendations stating what or how to eat, in relation to health and sustainability in Switzerland through previous work by sociologists [6]. The purpose of including this step is to use cross-disciplinary knowledge to link social perceptions and practices to evidence if such practices are or are not healthy and sustainable. Furthermore, this step can help ascertain which data are (or are not) available to assess perceptions that may have implications for policy making. “Prescriptions” (in this work, focused on perceptions of health and sustainability) can be issued by doctors, health authorities, popular magazines, retailers, or other entities, and can have various levels of scientific foundation. Work by Godin and Sahakian (2018) [6] found that relevant food prescriptions in Switzerland were a “balanced diet”, “local and seasonal” foods, “organic” foods, a “less and better meat” diet, “vegetarian and vegan” diets, and a “slimming diet” [6]. A “gluten free” diet was also identified as an emerging trend to be perceived as a healthy diet.

In order to match closely the existing perceptions for healthy and sustainable diets in Switzerland, we compared these prescriptions and emerging trends to the data available on self-reported practices related to “special diets” in the first Swiss national MenuCH nutrition survey [10] by the Swiss Federal Food Safety and Veterinary Office (FSVO). MenuCH is a data repository gathered by face-to-face or phone interview of self-reported consumption on one or two selected days within the years of 2014 or 2015 of 2085 people living in various regions of Switzerland. A total of 3860 individual responses were completed in the survey. Details of the survey methods are reported in MenuCH documentation [10] and peer-reviewed publications [13]. For more information on the treatment of MenuCH data, see Appendix A.

Previous work by Chatelan et al. (2017) [14] used MenuCH to investigate the proportion of the population with dietary practices that follows the Swiss Food Pyramid guidelines, and found that adherence is uniformly low across Switzerland [14]. We thereby did not assess diets conforming with the Swiss food pyramid prescription. Some of the identified prescriptions (i.e., “local and seasonal”; “organic”; “less and better meat”) were not possible to assess given the data available through the MenuCH survey and also due to the subjectivity of these prescriptions (i.e., how far of a radius qualifies as local? What quality indicators describe “better meat”?). From MenuCH, we were able to assess eating as a “vegetarian”, “vegan”, and “a slimming diet” based on self-identified adherence to “special diets.” We distinguished environmental impacts of more expensive cuts of meat to investigate prescriptions regarding less and better meat. Although it is not a predominate prescription, MenuCH data were available to assess self-declared “gluten free” diets as well. Note that MenuCH data had inconsistencies regarding declarations of special diets (e.g., vegan), and the actual foods reported (e.g., vegans declared animal product consumption).

Furthermore, because parallel sociological research by the authors suggests that in Switzerland education level (e.g., and not income) as well as gender may influence dietary choices, we also distinguished diets reported by people of three levels of education (primary, secondary, and tertiary), and by declared binary gender (men and women). These various categories are all referred to as “diets”. In summary, in this study the following diets were assessed for both men and women: Average, vegan, vegetarian, slimming, gluten-free and different education levels were also investigated. Results were sensitive to the type and quality (i.e., price) of meat as reported in the survey.

### 2.2. Screening GHG of Dietary Surveys

The average grams per day of each reported unique food name were calculated for each diet from MenuCH data. Life cycle inventory data, which are internationally recognized data used within LCA, were used as the basis to estimate the GHG of each food (i.e., from agricultural production and transformation/processing when relevant). Life cycle inventory data are a quantitative description of the inputs (e.g., energy, fertilizer) and outputs (e.g., GHG emissions) of agricultural processes [15]. A complete table of the matched MenuCH unique food names, the simplifying assumptions, the life cycle inventory entry chosen, the adjustment factors, and the GHG emission factor is available within Appendix A, where 99.8% of the mass consumed in the average diet was able to be matched to life cycle inventory entries. We did not assume differences in packaging, cooking, and consumer transport across diets (e.g., we had no evidence to base assumptions that men and women would purchase food packaged differently), and thus did not include life cycle inventory for these other life cycle processes.

The Quantis internal life cycle inventory database as of 2018 [16] and ecoinvent v3.4 [17] were the main life cycle inventory databases used. Other sources were complimented if the data were unavailable, i.e., for alcohol [18], and fish for which we used the newer version of ecoinvent v3.6 which became available to us during the course of this study. Furthermore for beef, we used an entry for a European dataset using the 2019 version of the Agri-footprint database [19] and corrected for the beef derived from dairy and suckling cow systems (70:30 ratio for Swiss beef). For all meat (chicken, pork, and beef) we used economic allocation factors linked to the various cuts of meat (e.g., fillet versus ground beef) reported in MenuCH. Building on knowledge of the Swiss beef market, we also applied an assumption that beef cuts with an economic allocation of greater than 1 (i.e., cuts with higher than average market price) were complimented by an import market causing the ratio between dairy and suckling cow to change to 50:50. In this study (as aligned with other previous work), beef derived from dairy cows had a lower impact per kilogram than beef derived from suckling cows. The rationale for using economic allocation factors to distinguish between cuts of meat to ensure results are sensitive to this market force were the following: First, meat tends to dominate the GHG impacts of diets in high income countries therefore more scrutiny on this category may provide further insight; second, sociological research [6] found eating “less and better meat” was a perception with respect to healthy and sustainable eating and the type of meat cut was the only parameter able to qualify “better meat”; and third, consumption of meat in high income countries like Switzerland tends to favor more expensive cuts which can drive increased production of meat due to a lower proportion of these cuts available per carcass.

There were two main methodological steps to link nutritional survey data to life cycle inventory. As described in more detail below: First, all of the food items within MenuCH were matched to life cycle inventory databases, and second, the discrepancy between the produced to consumable food amount was resolved (i.e., 100 g of dry pasta at a production facility was assumed to be equal to 250 g of cooked pasta). The life cycle inventory database entries were then run through SimaPro software to quantify the Global Warming Potential based on a 100 year time horizon (GWP100) in accordance with the IPCC 2013 AR5 report [20].

In MenuCH, 1519 unique food names were reported (in French, German, and Italian). These food names were matched to the life cycle inventory databases. Composite foods (e.g., ham croissant) were matched to a maximum of 3 database entries representing major ingredients using simplifying assumptions for proportions. It was not within the scope of this study to perform a full analysis of the food production balance in Switzerland (i.e., origin of imports and percent of consumed domestic production), and in any case life cycle inventory data availabilities to complement such an analysis are limited. Therefore, datasets were selected to represent European or global trade-based market averages, and not necessarily Swiss production. When there were multiple life cycle inventory entries that could be associated with a food name (e.g., if “bread” is the unique food name, corresponding life cycle inventory entries could be wheat grains, processed wheat grains, or bread), the most downstream life cycle inventory entry “closest to the consumer” was preferentially selected. Due to the data available and the inherent differences between how foods are consumed in practice, the system boundaries of every life cycle inventory entry was not the same (i.e., some foods like “bread” include processing, where as other foods like “lettuce” do not), but represented the best available life cycle inventory data to match the item being consumed. Due to data unavailability we did not attempt to account for missing data across the life cycle of food items, and in this way the reported environmental impacts should be seen as an indication of the greenhouse gas emissions related to producing food to supply the swiss diet, and not the entire “food system” impacts which include e.g., transformation, transport and logistical (storage), and cooking impacts. The implications of this methodological choice on the result magnitude are further discussed in the discussion and limitations section of the study.

After matching to life cycle inventory, the next step was to resolve the difference between consumed amounts with the corresponding produced amount. Losses across supply chains (e.g., spoilage), inedible food parts (i.e., the peels and bones), and moisture gains/losses from cooking require adjustments of the consumed food quantity to obtain the produced quantity. To adjust for losses (before retail) and wastes (at retail and by consumers) we used the supplementary data from Beretta et al. (2017) [21] on food loss and waste in Switzerland which includes “avoidable” wastes e.g., due to spoilage and “unavoidable” wastes e.g., due to processing residues [21]. If a life cycle inventory entry represented a product at the farm gate all losses and wastes across the supply chain were considered to adjust the value. If a life cycle inventory entry corresponded to a facility such as a mill or plant, only the avoidable fraction of losses (i.e., spoilage) was considered, as life cycle inventory entries for facilities already account for unavoidable losses (i.e., processing residues) between the farm and processing facility. Juices and any foods where the life cycle inventory was not available for the final product, but was available for a main ingredient, we applied basic adjustment factors (e.g., adjusting to a calorie ratio, or making assumptions such as 2 kg of apples to produce 1 L of apple juice). The complete list of the adjusting fractions applied for each food–life cycle inventory pair are available in spreadsheet form in the Appendix A. Foods were then compiled into categories in order to more easily visualize and interpret results. Yogurt and cheeses were put in the dairy food category whereas milk drinks were considered beverages. Red meat included beef, pork, lamb, and horse meat. Results were demonstrated as GHG per diets as they were consumed, and also as an adjusted 2000 kilocalorie-equivalent daily diet to normalize the difference between men and women (where women consumed less kilocalories overall). Results were also demonstrated per “average” kilogram and kilocalorie consumed in each diet.

### 2.3. Deriving a Relative Disease Burden Score for Screening

To investigate the relative magnitude of disease burden we used the Health and Nutritional Index (HENI), a disease burden-based nutritional assessment tool built on the work from the Global Burden of Disease study series [22]. HENI quantifies the marginal disease burden associated with the intake of 16 dietary factors that cover both food groups and nutrients, as well as for alcohol [11,12]. More specifically, we first provides dietary risk factors that quantify disease burden for adults (aged 25+) associated with a gram consumed from each of nine food and beverage groups (fruits, vegetables, nuts and seeds, whole grains, red meat, processed meat, milk, alcohol, sugar-sweetened beverages) and six nutrients (omega-3 from seafood, calcium, fiber, polyunsaturated fatty acids, saturated fatty acid, trans fat, and sodium) that have been recognized as dietary risks in the global burden of disease work. This led to a total of 18 dietary risk factors that could be applied. Any other potential diet-mediated health benefits of impacts beyond the evidence in the global burden of disease studies, e.g., related to non-seafood omega-3s, sugars from foods (not beverages), were not included in this study. We adapted the work by Stylianou et al. 2018 [11] to estimate age- and gender-adjusted marginal dietary risk factors specific to the Swiss population in 2016 by coupling risk ratios (RR) [22] with disease burden rates for the Swiss population in 2016 [23] and the corresponding population distribution. The Swiss-specific dietary risk factors are expressed in micro-Disability Adjusted Life Years (μDALY) per gram of food or nutrient and ultimately indicate the all-cause positive (avoided μDALY, e.g., negative dietary risk factor estimates) or detrimental health effects (positive dietary risk factors). Just like a micrometer represents 1/1,000,000 of a meter, a microDALY represents 1/1,000,000 of a year, which results that a μDALY is 0.53 min of healthy life lost, or approximately 30 s. These disease burden scores per gram of consumption are compatible with the LCA disease burden assessment framework [24]. Each dietary risk factor is available in Appendix A. The various diseases (e.g., cardiovascular disease) that result from each dietary risk factor are also available in Appendix A. When deriving the dietary risk factors care was taken to avoid double counting e.g., counting twice benefits or impacts that can occur for a nutrient and a food item that contains the nutrient. Due to the nature of epidemiological evidence being based on correlation, there remains a risk of double counting for nutrients such as saturated fat and products that contain saturated fat such as processed meat, although there is evidence that health impacts of processed meat are independent of saturated fat content [25]. Specifically, to avoid double counting we removed the benefits of calcium from milk (beverage) products which were accounted for separately in the “milk” category. Also to avoid double counting between fiber and fruits, vegetables, whole grains and legumes (f, v, w, l), we created a separate dietary risk factors for fiber f, v, w, l that does not include benefits of lowering heart disease risks which are already included in the benefits of consuming these whole plant foods.

We then calculate for each diet, the reported cumulative intake in each risk component (in e.g., g sodium per person per day) by multiplying the daily consumption of each food item (e.g., g of gruyere cheese per person per day) by its risk component profile (in e.g., g sodium per g of gruyere cheese) and summing up across all food items consumed in the considered diet. The result of this is available in Appendix A.

For each diet, we finally calculate the disease burden associated with each risk factor by multiplying each of the 18 dietary risk factors by the difference between the reported cumulative intakes and the corresponding theoretical minimum risk levels (TMRL) considered in the global burden of disease work. TMRLs indicate the intake level for each risk factor that theoretically leads to no increase in disease risk. TMRL provides minimum level of risk for both diets “low in” certain elements (meaning beyond the TMRL there is no additional benefit when consuming more) and for diets “high in” certain elements (where below the TMRL there is no incurred risk that can be quantified).

### 2.4. Determination of HENI-Relevant Food Groups and Nutrients Profile

To determine the HENI-relevant food groups and nutrients profile for each of the 1519 unique food names in MenuCH we first identified food groups corresponding to each food-based dietary risk factor using the subcategories listed directly in MenuCH (e.g., fruits). We then determined the nutrient profiles for each nutrient-based dietary risk factor using the default nutrition data per 100 g of each food provided by MenuCH. Lastly, for food groups and nutrients unavailable in MenuCH data (specifically whole grains and omega-3 from seafood) we matched the food names in MenuCH to the closest name in the United States (US) National Health and Nutrition Examination Survey (NHANES) [26] which includes a comprehensive US food and nutrition database. This matching resulted in >95% of the mass and dietary energy (kilocalories) consumed via the identified whole grain products to be accounted for (with <5% of products not having an identifiable match in the US NHANES database). Transfat was not reported in the MenuCH nutrient database, and thus our disease burden analysis did not consider the contributions from this risk. We expect that transfat would have negligible influence in the overall health performance of Swiss diets as there are significant reductions in the uses of transfat in Europe and Switzerland compared to the US [27].

## 3. Results

### 3.1. Consumption Results in Terms of Portion Size of Nutrients

Table 1 summarizes the total mass, dry mass (total mass corrected for water weight), and energy of food consumed daily. A description of the diets extracted from MenuCH, e.g., how many individual survey responses were obtained from self-identified vegans, is available in Appendix A. Given the small number of responses for certain diets, for example those self-identified as vegetarians (*n* = 158) and especially vegans (*n* = 15), the diets assessed in this study should only be seen as indicative and not as statistically significant or representative of what various sub-populations are consuming across Switzerland.

As for the composition of these diets (Figure 1A,B) the average Swiss diet is comprised more or less equally (each 15% of total food consumption on a mass basis) of fruits, vegetables, cereals, dairy products, and meat. Given the large influence of gender (due to both quantity and type of foods consumed), consumption results in Figure 1A,B are presented separately for men and women across all diets. Compared to the average diet, the average vegetarian consumed roughly the same amount of dairy, cereals (although higher proportion of whole grains), and eggs, and about 20% more fruits and vegetables, 15% more candies and sweets, and 230% more legumes (not including meat and dairy product substitutes). These findings using Swiss data, contradict assumptions made in previous studies (albeit not based on survey data) e.g., that in Europe vegetarians substitute meat with cereals [28]. Compared to the average diet, the average vegetarian also reported consuming substantially less bottled water, sugar sweetened beverages, coffee, and milk, more tap water, tea, and milk substitutes (e.g., rice milk), and similar quantities of juice and alcohol. Compared to the average diet, the average vegan (15 survey responses) consumed substantially higher amount of whole grains, fruits, vegetables, sauces, animal product substitutes, and soups. They also consumed more tap water, tea, milk substitutes, and less alcohol than the average Swiss diet (where all male vegans reported no alcohol consumption). Given the role of soups and beverages, results were also scaled to dry matter content (by adjusting by food-specific water content as reported in MenuCH), and to dietary energy consumed (according to food-specific MenuCH values). This demonstrated that although the vegan survey responses reported more beverages and soups than the average Swiss diet, the total amount of dry mass and dietary energy was also higher than average.

Our analysis revealed that self-identified diets may not be aligned in practice with what is generally recognized as the dietary prescription; for example, a subset of self-reported vegans (i.e., animal product-free diets) and vegetarians (i.e., meat-free diets) reported consuming dairy and meat respectively. Sixteen out of the 104 surveyed vegetarians reported eating meat leading to an average of 11 g per person per day (g/p/d). One self-identified vegan participant reported eating 56 g of processed meat, which led to a group daily average of 4 g/p/d, and 13 out of 15 vegan survey responses reported dairy consumption leading to an average of 40 g/p/d. It is unclear if these practices represent an issue or misunderstanding with the survey or flexibility between self-declared dietary preferences and actual practices. Furthermore, other observed differences in vegan and vegetarian diets (e.g., more whole grains, less bottled water) suggest that the actual practices for these diets may be tied to lifestyle choices that are not restricted to only consumption of animal products.

Slimming diets contained more animal products than average, while gluten-free diets contained more fruits and vegetables than average and overall less mass, dry mass, and dietary energy than slimming or average diets. Total consumptions slightly increased with education level, with slightly higher intakes of legumes, nuts and seeds, whole grains, and dairy products such as yogurt and cheese (but less milk). Analysis of dietary patterns by gender showed that men consumed nearly double the amount of meats and cereals compared to women. Men also reported drinking more sugary sweetened beverages and alcohol whereas women drank more teas and herbal infusions. Men of higher education levels also tended to eat more expensive cuts of meat (e.g., steak), for example with the secondary education level actually consuming slightly less red meat than primary education level, but higher proportion of more expensive cuts (Appendix A). We found contrasting patterns in meat consumption between women of higher education levels (that consumed less meat) and men of higher education levels (that consumed more meat).

### 3.2. GHG of Swiss Diets

GHG results are summarized in Table 2. The average Swiss diet (combining men and women) was associated with 2.1 tons of CO_2_e per person per year (tCO_2_e/p/y) with 1.7 tCO_2_e/p/y for food items and 0.4 tCO_2_e/p/y for beverages. For the average diet, red meat and dairy (solid and liquid) were each 20% of the total GHG footprint, with other meats (including fish) contributed another 15%, meaning animal products contributed to about half of the average diet GHG footprint. Other significant contributors were cereal products (10%), and alcohol (10%). The total carbon footprint of all diets and the relative contribution of various food and beverages are depicted in Figure 2A,B. These results are adjusted by production to consumption factors that include waste across supply chains. Results were generated also without the waste adjustment factors (Appendix A). These results suggest that the impact of the average Swiss diet goes up by about 20% when including adjustment for food waste, and that vegan and vegetarian diets are even more sensitive to waste factors because they have a higher proportion of foods with higher wastage at retail and household level (e.g., cereals, legumes, fruits, vegetables). 

The diets with the lowest carbon footprint were the average vegan (1.3 tCO_2_e/p/y) and vegetarian (1.4 tCO_2_e/p/y). Vegan and vegetarian diets had similar associated GHG, however explainable by different types and quantities of food consumed.

Given there were variations in the total amount of food and dietary energy consumed, for example across men and women, we also present GHG results (Table 2) scaled to mass and to a 2000 kilocalorie diet. The largest differences in all cases was due to animal products consumption, particularly red and processed meat. Vegan diets generated nearly half the carbon footprint than the average diet per either gram or kilocalorie of food consumed, with the vegetarian diet falling in between. For beverages, impacts per mL on average were similar across diet types, however in total the minimum and maximum were roughly a factor 2× with the lowest GHG associated with beverages consumed by vegan men due to no reported alcohol consumption by any of the 5 respondents (results are not shown, and are not statistically robust given the small amount of suveys).

The impacts of diets reported by men (1.7–2.6 tCO_2_e/p/y) had a larger range than diets reported by women (1.1–1.7 tCO_2_e/p/y), where for diets other than vegetarian and vegan for women had a more restricted range from 1.6–1.7 tCO_2_e/p/y whereas for men these same diets ranged from 1.9–2.5 tCO_2_e/p/y. The larger range observed in the male diets was largely sensitive to the amount and type of meat consumed, where for example the men in the secondary education group consumer larger amounts of more expensive meat cuts leading to a higher impact per kilogram of meat than in the primary education level. Beverages also played a role, where for example none of the male vegans (5 survey responsents thereby not statistically robust) reported consuming alcohol, leading to a substantially lower footprint for the beverages consumed by male vegans.

To identify priorities across the population we also ranked greenhouse gas impacts in comparison to quantity consumed in Figure 3. The interpretation of this figure is that from a GHG perspective red meat, dairy, processed meat, cereals, and alcohol are food categories that have a high contribution to the overall GHG impact of the Swiss diet as combination of their consumed amount and their per kilogram impact.

### 3.3. Disease Burden Scores of Diets

In addition to the averages combining men and women summarized in Table 2, Figure 4 presents the disease burden scores for each diet separated by men and women, differentiating the contribution of each risk factor (e.g., diets low in whole grains). We demonstrate the results as cumulative benefits (decreased risks) and risks related to diets as the sum of each of the 18 dietary risks. This depiction is correct in the direction (i.e., if positive or negative) and comparable across diets and to each risk factor independently; however the overall magnitude of impact or benefits are likely underestimated in magnitude as health benefits and impacts towards the same disease outcome may be multiplicative in their effect and not additive. These scores are in comparison to global burden of disease reference diet corresponding to the amount with the lowest theoretical risk (i.e., 0 disease burden) for each risk factor, thus they represent the “potential room for improvement”, or distance to best diet as defined by the global burden of disease. The disease burden score for risk factors corresponds to consumptions below the amount that lead to no risk (for any risk factor where the diet is “low” in) and to consumptions above the amount that causes disease burden (for any risk factor where the diet is “high” in). Overall, the vegan diet led to the lowest disease burden scores (71 μDALY/p/d), mostly due to higher reported consumption of whole grain products that are beneficial for health, and the low consumption of processed meats that are detrimental for health. The vegan diet, however, had higher sodium intake than the other diets corresponding to 14 μDALY/p/d), where soups, soy sauce, and pasta dishes were the most important contributors. Results on vegan diets are based on 15 respondents are therefore not statistically robust to represent the population. The vegetarian diet also generated substantially lower disease burden scores compared to the other diets, mostly due to the lower amount of processed meat compared to average and also due to more fruits, more nuts and seeds, and less sugar sweetened beverages. The vegan and vegetarian diets also lacked benefit (albeit a small difference) from omega 3 fatty acid from seafood (notably any benefits from omega 3 fatty acid from non-seafood sources were not included, as evidence is unavailable in the global burden of disease studies).

Diets reported by men had the highest disease burden scores (with processed meat having the highest contribution to this score). Processed meat consumption by the average man was associated with an increased risk of 24 μDALY/p/d which can be interpreted as approximately 13 min of healthy life lost each day; increased risks were then followed by low whole grains, low nuts and seeds, and high alcohol. The associated health benefits with eating fruit and vegetables, whole grains, legumes, etc. were similar among men and women, where women consumed only slightly more beneficial foods than men overall. When adjusting by the total quantity of kilocalories consumed (data not shown) there is nearly no difference between the disease burden scores of men and women, 0.054 and 0.055 μDALY/kcal consume respectively, and the largest benefit is seen for vegans having only 0.024 μDALY/kcal consumed.

The magnitude of possible improvement between the average diet and global burden of disease reference diet (assuming all risk factors are at the theoretical minimum, zero risk) was ranked for each food category (Figure 5) according to the population average. This ranking shows a prioritization of dietary shifts according to their potential benefit if shifting from the average diet to the global burden of disease reference diet (lower figure), and the size of the dietary shift required (upper figure). Shifting towards diets high in whole grains (maximum benefit of reducing about 20 μDALY/p/d) and low in processed meat (maximum benefit of reducing about 14 μDALY/p/d) could offer the highest possible health benefits for single dietary elements. Increasing the intake of nuts and seeds, vegetables, fruits, legumes, fibers (from sources other than fruit, vegetable, whole grains and legumes), polyunsaturated fatty acids and seafood omega 3s also offer benefits, as well as decreasing alcohol, saturated fatty acids, sodium and sugary sweetened beverages. This figure suggests reducing red meat and increasing calcium, milk are low priority for reducing disease burden at the level of the population. In this study we add the disease burden scores to indicate totals, however the measurable disease burden benefits of improving each dietary risk factor is multiplicative and not additive with respect to a total disease burden [22], (which means that the cumulative benefit from all shifts is lower than the sum of benefits of each shift individually because there are interactions between the risk factors).

## 4. Discussion and Limitations to the Study

The novel contribution of this study is providing an analysis of self-reported practices in a given population to identify priority (and non-priority) policies that can simultaneously improve health and GHG emission. Similar work has been performed recently, for example for the United States [29]. A key novelty in our study is the separation of diets by reported gender and the use of the HENI [11] metric to derive health scores from global burden of disease. Unlike nutritional indices or qualitative indication of a food’s “healthiness”, HENI disease burden metrics help prioritize dietary shifts related to the relative order of magnitude of their benefit (or impact) on health for a given population. This study provided evidence that focusing on key food groups and men provides opportunities for priority win–win interventions. Mainly, this work identified the key win–win priorities (when interpreting in combination Figure 3 and Figure 5) as decreasing processed meat and alcohol consumption especially for men of higher education levels. Another identified priority is to increase whole grain consumption, e.g., to replace refined grains, across the population (including for people with diets low in animal products), yet this intervention would have less importance for reducing GHG emissions. This study also provided evidence that decreasing sugary sweetened beverages can provide win–win benefits for health and likely greenhouse gas reduction (i.e., if replaced with tap water), albeit is a much lower priority in comparison to other dietary shifts for Switzerland. Reducing red meat and dairy have key GHG benefits (depending on what may be substituted), yet according to the global burden of disease work have less importance for health in comparison to other components of the diet (see Figure 5). These findings suggest that policies towards dietary shifts to reduce red meat (especially for men of higher education) and reduce dairy for all individuals are priority for GHG emission that offer little trade-offs in the context of health—yet overall GHG emission and health impact would be dependent on what items are substituted. The impacts or benefits of what substitutes items that are decreased in the diet were not specifically assessed in this study, although other studies have shown benefits for plant-based substitutions (legumes, nuts and seeds, and whole grains) [1].

The priority levers of change identified in this study (e.g., decreasing processed meat and alcohol for men and increasing whole grains for all) are not key aspects of the current discourse at various stakeholder and scientific events attended by the authors or the studied “pro” or “no” meat public discourse in Switzerland which tends towards specific moral and emotional messaging. Generally discourse around food tends to focus on processed foods, and “pro” or “no” meat consumption in general. Emotional messaging and debate regarding meat (and animal product) consumption in Switzerland tends to use associations of national pride to promote meat and cheese, or use imagery leading to disgust to denounce animal husbandry [30]. Further study could consider how such emotional messaging may (or may not) change practice in comparison to other interventions such as food availability in relation to daily mobility and in schools, and beyond. Investigation of emotional discourse could also shed light if the communication of health impacts (such as those found in this study) and emotions that may elicit is or is not helpful in changing dietary choices.

In this study we first related sociological findings on predominate perceptions of healthy and sustainable diets (referred to as “prescriptions”) in Switzerland [6] to the dietary survey data available through the first Swiss National Survey, MenuCH. Investigation of the sociological context for the perception of health and sustainability combined with impact assessment is a developing area of inter-disciplinary work that can provide multiple layers of insight for policy and change making [31,32]. An initial finding from addressing the MenuCH data from a sociological perspective is that quantitative data are largely unavailable to assess the qualitative perceptions of healthy and sustainable diets. A clear example is the association of eating “balanced” or “locally” with health and sustainability; data were not available through menuCH (or otherwise) to assess the healthiness and sustainability of actual practices of people who aim to eat a “balanced” diet or “locally.” Other work suggests that Swiss in general (regardless of general knowledge of the Swiss food pyramid) are not consuming diets aligned with the food pyramid balance [10,13], and that “local” eating provides limited benefit for GHG reduction [33]. Another key findings from considering the sociological perspective is that vegetarian and vegan diets are perceived as healthier and more sustainable and were also found to have better GHG and health scores. Nevertheless, especially the health scores could be improved by food consumptions that are outside of these dietary prescriptions (i.e., increasing whole grains). Also the observed benefits of vegan and vegetarian diets were driven by key aspects of consumption i.e., consuming less processed meat) that could be practiced outside of this dietary prescription (e.g., in flexitarian or meat reduction diets, or simply diets that focus on eating at home as (data not shown) the majority of meat consumption was out-of-home. These finding suggests that perceptions that have high traction in day-to-day society with respect to health and sustainability may not be aligned with the evidence for specific win–win policies.

In relation to the first scientific question we aimed to address about the comparative magnitudes of health and environmental impacts, we found that vegan and vegetarian men had the lowest disease burden scores, and men with secondary education had the highest disease burden scores (Figure 4). Given the average diet reported by men was also associated with the highest level of GHG, these findings provide evidence that targeted policies for dietary shifts specific to men (e.g., to consume less overall, and/or to limit processed meat and alcohol consumption) could offer a “win–win” impact reduction for both health and GHG. There is building evidence that it is important to analyze differences between men and women in the context of dietary change [29]. Notably, advertisements promoting meat consumption for men have been a recent controversy in Swiss media [34].

For our second research question on most influential factors, disease burden scores for vegetarians, vegans, gluten-free, and other special diets were largely influenced by whole grains, nuts and seeds, processed meat, and alcohol. This finding, along with findings regarding differential use of bottled water between special diets (i.e., vegetarians consumed less bottled water), suggests that there may be additional consumption choices important for health and/or sustainability performed by people self-declaring special diets, but that are beyond the definition of the special diet. Future study could be expanded to understand what drives this larger set of practices or lifestyle choices for example to also consider people’s mobility or social interactions.

As our results are based on the robustness of the global burden of disease study and epidemiological evidence, which is evolving, the results of this study would change if there are changes in the global burden of disease results due to new evidence or interpretation of evidence. (This is also true for the environmental impact data, although that tends to be more robust than nutritional epidemiology.) The global burden of disease work does not include risk factors where there is a lack of conclusive epidemiological evidence e.g., there is insufficient evidence to consider risks from general sugar consumption (apart from beverages) [35], as well as omega 3 sources outside of seafood. There are recent studies suggesting that the global burden of disease factors for saturated fat may be overestimated [36], and thus risks due to saturated fat may be overestimated in this study.

A limitation of using disease burden metrics is a lack of insight on nutritional adequacy. As a parallel analysis we also checked quantities of key nutrients that have been raised in other studies as potentially inadequate in vegan and vegetarian diets, specifically protein and vitamins B2 and B12 [2]. The average of the considered diets ranged from 64–98 g of protein per day, with female vegans and vegetarians consuming the least 56 g protein/d, and male vegans consuming more than the average male (113 g protein/d versus 98 g protein/d) due to high consumption of legumes, nuts and seeds, and animal product alternatives. Average requirements of protein according to the European Food Safety Authority (EFSA) for adults are 0.66 g protein/kg of body weight, or about 40–52 g protein/d given 60–80 kg bodyweight, suggesting all diets (but not necessarily all individuals) had adequate protein intake [37]. We did not investigate in this study specific amino acid consumptions (as these data were not available in MenuCH) which is an area of important research when considering protein sufficiency for various diets. As for vitamin B2 and B12, all diets were below the average requirement suggested by EFSA as well as European averages; this seems to be related to the quantity of B2 and B12 reported in the MenuCH database for milk, e.g., listed as 0.16 mg of B2/100 g and containing no B12—which is contradicted in other references [38]. These findings suggest that further scrutiny is needed to enable the MenuCH data help investigate nutritional adequacy.

Screening the comparative magnitudes of GHG demonstrated that vegan, vegetarian, and gluten-free diets all had similar low impact profiles (albeit for different reasons), and the average diet reported by men having the highest GHG. Based on 15 respondents, vegan diets had the lowest GHG emission and disease burden scores per gram and per kilocalorie, and this finding held true even given higher amounts of losses and wastes in supply chains related to the products consumed in these diets. We were unable to assess differences in household food waste between different diets which could be substantial given the finding that the prescriptions may be associated with other lifestyle choices. One unexplainable finding was that the vegans according to the MenuCH data reported to consume more mass and kilocalories than other diets. Possible explanations could be a statistical anomaly due to low sample size (although we found this consistently across men and women with no individual outliers), other lifestyle choices such as increased physical activity, or physiological aspects e.g., feeling of satiation. It was not the goal of this study to do a robust statistical analysis to understand the margin of error on the Swiss MenuCH survey results which has been done in other studies [13] albeit not for special diets. Further research is needed with targeted sampling on special diet (e.g., for vegans and vegetarians from each gender and language region) in order to have statistically significant results and inform actual practices related to these diets.

A limitation of the GHG assessment of vegan and vegetarian diets is related to the assessment of meat and milk substitutions. Without specific data on these substitutions, when not otherwise indicated (e.g., rice milk) they were modeled as soy, with a relatively high impact for a plant-based food with 6 kg CO_2_e/kg soy product consumed due to losses and wastes across the supply chain (Appendix A). This value is aligned with more in-depth study of meat replacement products with various ingredients [39], however the results for the meat and dairy replacements should be interpreted with caution as these products can vary widely. On average, vegans drank more juice, coffee, and milk drink substitutes than vegetarians. The impact of beverages is highly sensitive to the dilution and concentration factors which range greatly from product-to-product and person-to-person (e.g., use of coffee pods or French press)—data which were unknown in this study. Another unexplainable finding was that reported gluten-free diets had similar emissions to vegetarian and vegan diets due to very low reported beverage intake, and not due to lower animal product consumption. Taken together, the general findings in this study suggest that meat and dairy substitutes as well as beverages can play a key role in the impacts associated with special diets. Furthermore, out of scope of this study but important for policy and decision making, other impact categories such as water scarcity and ecotoxicity are also important to consider for legumes and nuts and seeds. These indicators are more difficult to assess than GHG at the diet-level and more meaningful to assess per product according to specific location of production [40,41,42].

Dairy and beef have high importance historically and currently in Switzerland. Dairy and milk were substantial drivers of GHG emissions, even for self-reported Swiss vegetarian diets and vegans (although consuming dairy is not aligned with the widely accepted definition of vegan). In our study further increasing dairy consumption was not found to be a major lever to improve health based on the current evidence used to create global burden of disease risk factors related to dairy and also due to the relatively high consumption of dairy in Switzerland. This finding is in contrast to media reports that the data from MenuCH show a need to increase dairy consumption in Switzerland to improve health metrics [43]. Beef products were also major drivers of diet-related GHG and especially for men with secondary education who consumed higher amounts of expensive meat cuts (e.g., steak). The finding from sociological research that Swiss identify with eating “less and better meat” as “more healthy and sustainable” is difficult to quantify—for example consumers seemed to associate “better meat” as Swiss (not foreign) meat and potentially “organic”, yet at the same time there is demand for expensive (“better”) meat cuts in Switzerland. Given this consumer demand, one way to quantify “better meat” would be using the price of the meat cut (more expensive is higher quality and thus “better”); however this is contradictory to the sentiment that “better meat” is less impacting, as economic allocation and consequential thinking suggests that more expensive cuts can drive production. Furthermore, there are trade-offs and inconsistent evidence suggesting organic and pasture-based (“better”) beef and dairy production may have lower feed conversion efficiency and thus higher GHG emissions per kilogram of milk or meat; yet these systems have other benefits such as nutritional quality of the milk, animal welfare, and biodiversity [44,45,46]. This suggests that GHG is not the appropriate indicator to assess the benefits of “better meat” if this practice is describing consuming organic and/or pasture-based beef. Finally, although not possible to quantify in this study, there may be quantifiable GHG benefits of consuming “better meat” if this perception is leading to practices that avoid meat associated with deforestation (i.e., due to imported feed) and consuming a variety of cuts instead of more expensive ones (e.g., filet).

As for education levels, a recent US study suggested people with higher education have substantially higher diet-related GHG [47]. In our study we found this was true for men with higher education levels due to purchasing a higher amount of expensive meat cuts which due to economic allocation have a higher GHG impact; however, we did not find the same pattern in women which had relatively stable impacts due to food consumption across education levels. We did find higher education levels tended to have higher impacts related to beverages, e.g., alcohol for both genders. Thus, we did not find strong evidence that food policies in Switzerland should target education levels specifically.

The overall magnitudes of GHG of all the diets are well-aligned with previous findings for average and special diets in other European countries [8]. As the purpose of this study was to compare across diets with a focus on impacts related to agricultural production, we did not include a detailed study of all life cycle processes such processing, packaging, refrigeration, preparation and washing (although some life cycle impacts were included for final products such as bottled water and bread). Without data available on the full value chain of food items, we assumed such additional impacts would be similar across diets from an environmental perspective e.g., vegans and the average Swiss would have similar practices related to food processing, packaging, refrigeration, cooking, and washing. If included these impacts would likely increase the footprint of the average diet by about 1.5 times [28,48], adding about another 1 tCO_2_e/p/y. The differences across diets for life cycle processes such as food processing, refrigeration, cooking, and packaging related to various consumer practices could be interesting to investigate in a separate study.

In all, especially given this study represents the impacts of food production and not all related dietary impacts (e.g., preparation at home), this study demonstrated that all diets, including diets with restricted animal product consumption (vegan and vegetarian), would still far exceed the limit of 0.6 tCO_2_e/p/y suggested by the federal office for the environment in Switzerland [49], as well as the limit of 1 to 3 tCO_2_e/p/y suggested by other sources reach the sustainable development goal of not exceeding a 1.5 degree temperature rise by the year 2030. These targets include all forms of consumption, such as mobility, housing, and material goods, and food [2,5,6]. Thereby, when it comes to decreasing GHG associated with the Swiss diet to be within planetary boundaries [50], shifting diets is just one of several other key interventions such as efficiency improvements (reducing food waste, and improving fertilizer efficiency) and reducing fossil fuel use [51]. If reducing food waste is accompanied by a decrease in the required food production to supply and feed Switzerland, reducing food waste (Appendix A vs. Figure 2) could lead to an emission reduction by about 20% (depending on the diet)—which is a non-negligible contribution to the GHG emissions associated with Swiss diets.

This study was also able to identify areas where the available survey data were not aligned with the data needed to assess healthy and sustainable eating in Switzerland. Specifically, there were insufficient data on the location and mode of production of the foods consumed in order to make any claims about organic or local diets. Including such data in a dietary survey may not be practical and pose additional burden on the participants and surveyors, thus identifying priorities related to specific to mode and location of production, such as biodiversity loss, should likely be targeted by research that is not dependent on survey data. As a separate finding, there were no data regarding the contents of whole grain and omega-3 fatty acids within the products listed in the MenuCH survey, which are key dietary risk factors. In order to obtain the consumptions related to these risk factors, we needed to match MenuCH to a secondary source introducing substantial workload. A recommendation would be to include omega-3 fatty acids and most importantly whole grains in future dietary surveys.

The results in this study are intended to be interpreted as screening values to indicate generally beneficial directions or “win–win” situations for national policy advice on healthy and environmentally sustainable diets, and should not be used as individual medical advice (which depends on e.g., hereditary factors, other lifestyle choices such as physical activity). Sources of uncertainty include the Swiss national survey (MenuCH) data which are reported food consumption from memory recall (not actual food consumption) [13,52], and the application of the epidemiological evidence derived from the global burden of disease study [53] which was used to derive the HENI (disease burden) scores. Other uncertainties include matching to life cycle inventory data, and specifying dilution factors for beverages (e.g., quantity of coffee beans per cup of coffee), which were particularly important for dietary prescriptions which results suggested had high variability in reported beverage consumption.

## 5. Conclusions

As the first study to screen the first Swiss national survey (MenuCH) in relation to comparative health and environmental metrics, we contribute to a novel methodological approach and findings relevant for future research and policy-makers interested in transitions to healthy and sustainable diets. The methodological approach started with sociological research on prescriptions, or guidelines for conduct around “healthy and sustainable” food consumption in Switzerland. We then related food consumption in the MenuCH survey to relative health and environmental metrics.

The key findings are: First, and similarly to other research on European diets, all Swiss diets (including vegetarians and vegans) are far above the suggested 0.6 tonnes CO_2_e capita per year sustainability target [49], which is a desired target for all consumption categories, including food and also transport and heating. Thus, even if the Swiss population were to eliminate animal products from their diets, the footprint is still above suggested per capita targets. Beyond dietary shifts and inciting changes in consumption practices, other levers of change include reducing fossil fuels, chemical fertilizers, and food waste in food provisioning systems, towards transforming food production on a systemic level. Second, we found that consuming more whole grains has the potential to provide the highest health benefit in Switzerland with little GHG consequence, and eating less processed meat offers the second highest health benefit also with high benefits for GHG. Reductions in alcohol consumption are also a win–win for environmental and health outcomes. Lastly, a gendered reading of the results indicates that supporting interventions towards more healthy and sustainable diets in Switzerland should include a focus on men of higher education levels, specifically in reducing alcohol and processed meat consumption.

In relation to further study, we recognize limits of the MenuCH data in relation to accounting for consumer perceptions and practices regarding healthy and sustainable diets. In particular, “local” diets which are often seen as a proxy for healthy and sustainable diets in Switzerland, were not able to be assessed although general evidence to date have not demonstrated that eating “locally” is especially material for GHG and health. Finally, MenuCH is a snapshot in time of current diets and may not be able to tell us much about people’s actions when they shift diets; a longitudinal approach could shed more light on practices and how they may change.

A “win–win” policy recommendation would be to increase what is both healthy and less impacting in diets at the expense of what is not healthy and more impacting, with a focus on men’s diets and decreasing processed meat and alcohol, while increasing whole grains and nuts and seeds. Decreasing dairy and red meat (especially for men) also offer a GHG benefit with a relatively small influence on health scores. Given the complexity of food consumption and its interrelation with other aspects of society, a food policy forum in Switzerland could help engage discussions on how to implement policies and interventions, for example in relation to other dimensions of everyday life, such as mobility, the world of work, and eating out.

## Figures and Tables

**Figure 1 nutrients-12-02745-f001:**
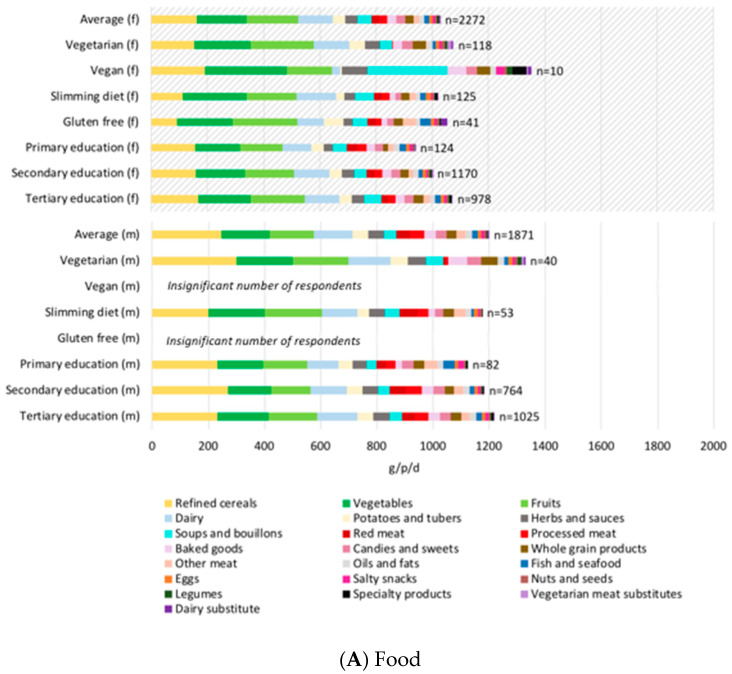
Swiss dietary consumptions for (**A**) food and (**B**) beverage categories; series (food categories) are ordered from most to the least consumed in the average diet; upper charts with diagonal shading are results for women (female, f), and lower charts results for men (m). The number of survey responses (“*n* =”) for each diet-gender combination is demonstrated on the graph. Results for male vegans and gluten free diets had less than 10 survey respondents and therefore statistically weak and not shown for comparison. Results for female vegans should be interpreted with caution due to the low respondent amount of *n* = 10. Source: Office federal de la sécurité alimentaire et des affaire vétérinaires: enquête national sur l’alimentation menuCH 2014/2015.

**Figure 2 nutrients-12-02745-f002:**
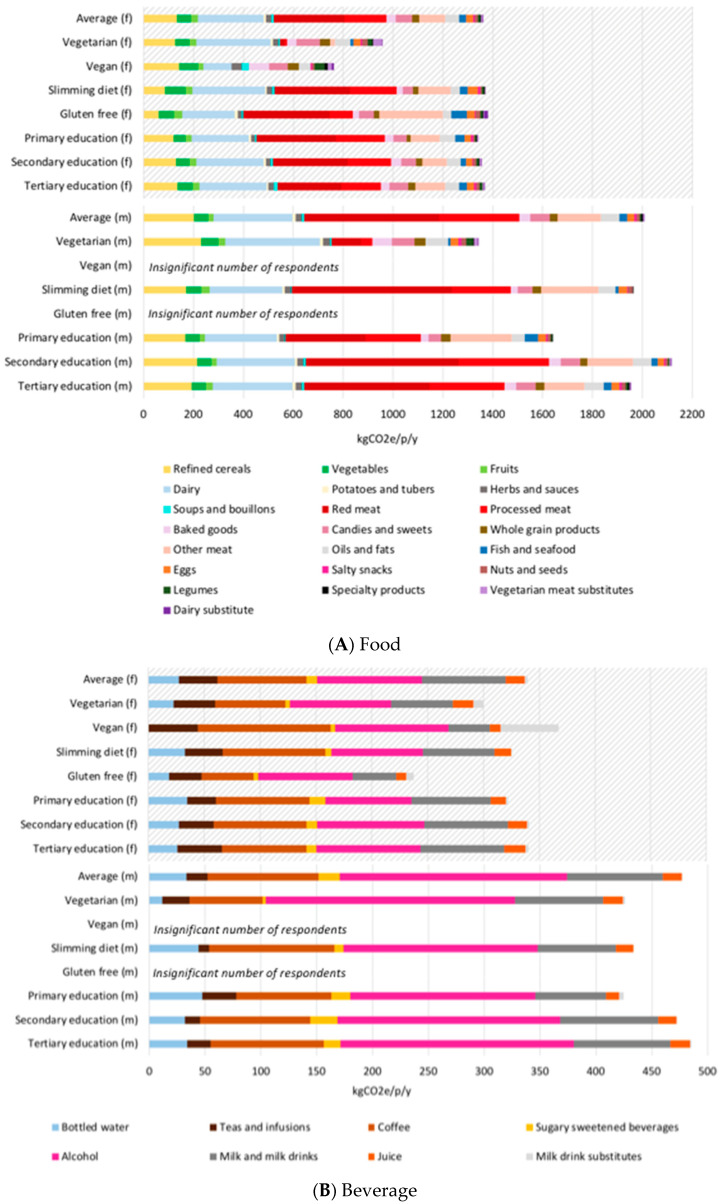
Total GHG emissions for Swiss diets for (**A**) food and (**B**) beverage categories; note the different scale between the figures; series are ordered from most consumed to the least consumed category in the average diet as shown in Figure 1A,B. Upper charts with diagonal shading are results for women (female, f), and lower charts results for men (m). Results for male vegans and gluten free diets had less than 10 survey respondents and therefore statistically weak and not shown for comparison. Results for female vegans should be interpreted with caution due to the low respondent amount of n = 10.

**Figure 3 nutrients-12-02745-f003:**
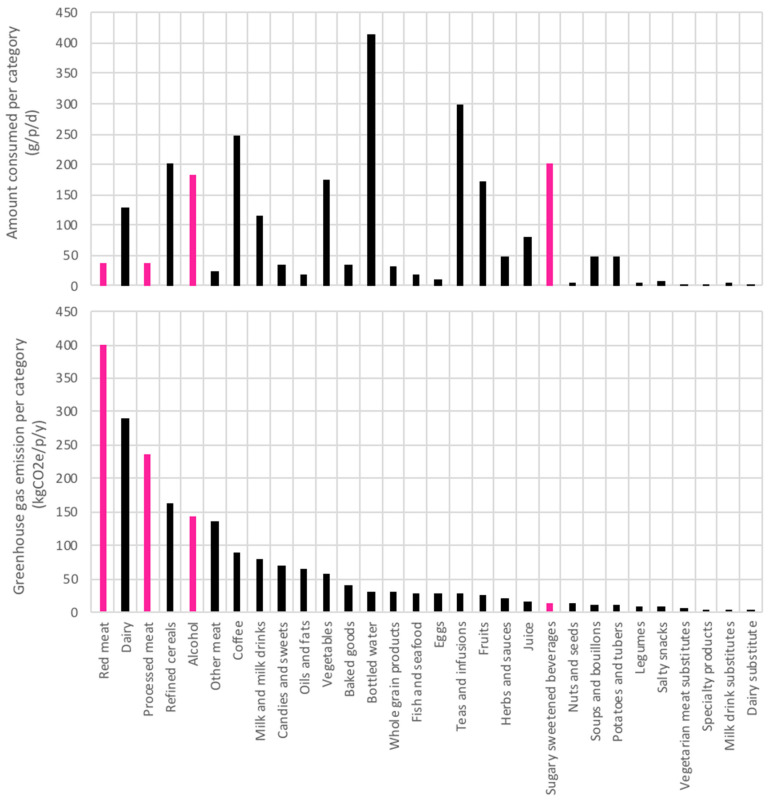
Total GHG per person per year per food category ranked from highest to lowest and associated quantity consumed as grams per person per day (g/p/d). Food categories in pink are also associated with a detrimental health impact score.

**Figure 4 nutrients-12-02745-f004:**
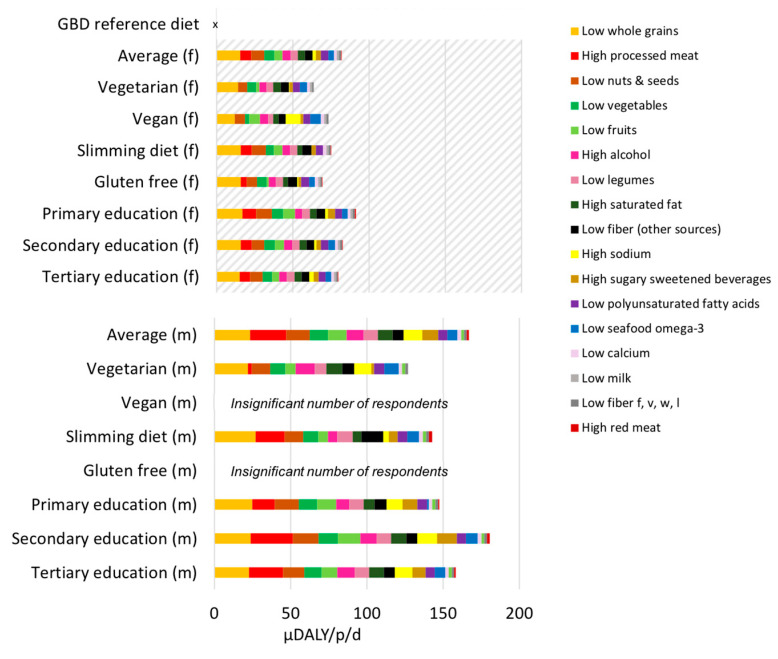
Disease burden scores in micro (μ) disability adjusted life years (DALY) per person per day (where 1 μDALY is about equal to 30 s of life lost). Risk factors are shown in color and ordered starting from the maximum contributor to disease burden (low whole grains) to minimum disease burden (high red meat) in the average diet (men and women). The demonstrated scores are in relation to the global burden of disease (GBD) reference diet which has a score of zero, to be interpreted that there is zero incurred disease risk when each risk factor is consumed at its theoretical minimum risk level (TMRL). The category “Fiber f, v, w, l” indicates fibers from fruits, vegetables, whole grains, and legumes, and “fibers (other sources)” from sources other than those. Note there were no data available on transfat consumption, so results are not shown. For risk factors listed as “high”, e.g., high processed meat, the interpretation is that disease risk is incurred due to consuming greater than the global burden of disease reference diet; for risk factors corresponding to diets “low” in an element e.g., “low whole grain”, the interpretation is that disease risk is incurred due to not consuming as much as the global burden of disease reference diet. Upper chart with diagonal shading presents results for women (female, f), and lower chart results for men (m).

**Figure 5 nutrients-12-02745-f005:**
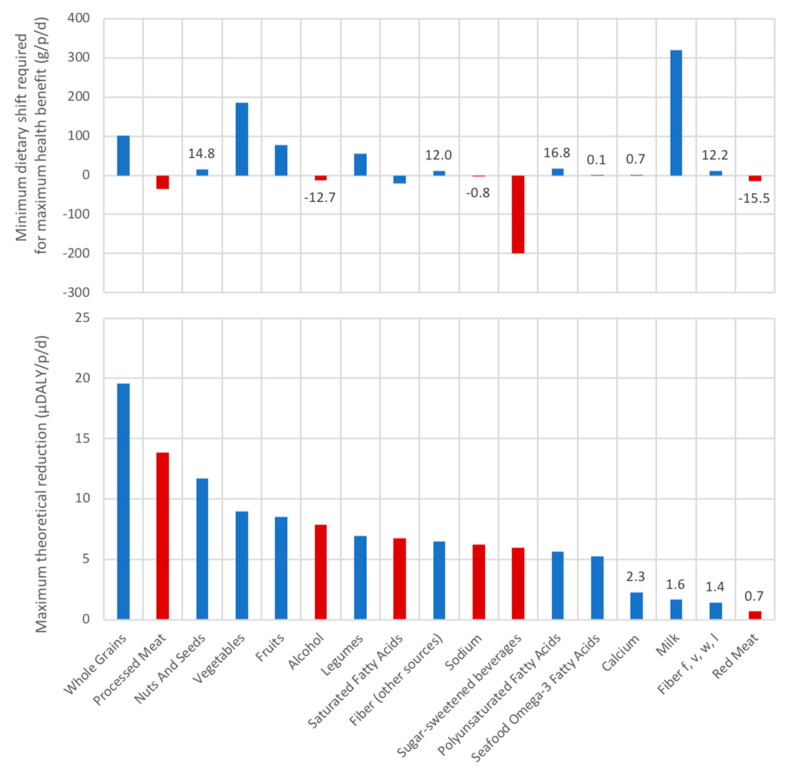
The maximum theoretical reduction in disease burden (μDALY/p/d) in priority order (lower figure); corresponding dietary shift (g/p/d) as the difference between Swiss average consumption and theoretical minimum risk level (TMRL). The category “Fiber f, v, w, l” indicates fibers from fruits, vegetables, whole grains, and legumes, and “fibers (other sources)” from sources other than those. Note there were no data available on transfat consumption, so results are not shown. Red bars indicate foods where consumption needs to decrease, and blue bars were consumption needs to increase in order to reach the global burden of disease reference diet with zero risk, i.e., the TMRL. The seafood omega-3 dietary shift corresponds to increasing fish consumption by 50 g/p/d given the average fish consumption in Switzerland and its omega-3 content. The alcohol dietary shift corresponds to decreasing about 1 dL/p/d of wine at 12% alcohol, or 2 dL of beer at 6% alcohol.

**Table 1 nutrients-12-02745-t001:** Results for mass (kg) and volume (L), dry mass (g), and dietary energy (kcal) consumed per person (p) per day (d) for various diets and split between food and beverage.

	Food			Beverage		
	Mass	Dry Mass	Energy	Volume	Dry Mass	Energy
	kg/p/d	g/p/d	kcal/p/d	L/p/d	g/p/d	kcal/p/d
**Average**	**1.1**	**384**	**1867**	**2.3**	**62**	**311**
**Vegetarian**	1.1	382	1855	2.4	47	249
**Vegan ***	1.5	481	2338	3.0	60	415
**Slimming**	1.1	329	1597	2.3	45	217
**Gluten free**	1.0	306	1497	2.0	38	184
**Primary education**	1.0	353	1716	2.2	58	286
**Secondary education**	1.1	375	1824	2.3	62	311
**Tertiary education**	1.1	396	1925	2.4	62	313
**Women**	1.0	389	1643	2.3	50	255
**Men**	1.2	439	2140	2.4	76	378

* Results are based on only 15 respondents—thereby not statistically robust at a population-level.

**Table 2 nutrients-12-02745-t002:** Results for the disease burden score in micro disability adjusted life years per day (μDALY/day), greenhouse gas emissions (GHG) in carbon dioxide equivalent (CO_2_e) for Swiss diets including; with total tonnes per person per year (tCO_2_e/p/y) as well as the total scaled to a 2000 kilocalorie diet per day, kilogram CO_2_e per kilogram of food (kgCO_2_/kg) (total food not dry mass), gram CO_2_e per kilocalorie of food (gCO_2_e/kcal), and kg CO_2_e per liter of beverage (kgCO_2_e/L). Beverage consumption includes tap water. μDALY can be interpreted as about 30 s of healthy life lost.

	Disease Burden Score	Total GHG	Total GHG Scaled for 2000 Kcal Day	Food			Beverage	
	μDALY/p/d	tCO_2_e/p/y	tCO_2_e/p/y	tCO_2_e/p/y	kgCO_2_e/kg	gCO_2_e/kcal	tCO_2_e/p/y	kgCO_2_e/L
**Average**	**120**	**2.1**	**1.8**	**1.7**	**4.1**	**2.3**	**0.4**	**0.5**
**Vegetarian**	88	1.4	1.1	1.1	2.5	1.6	0.3	0.4
**Vegan ***	69	1.3	0.9	1.0	1.8	1.2	0.3	0.3
**Slimming**	99	1.9	1.9	1.5	4.0	2.5	0.4	0.4
**Gluten free**	98	1.7	1.9	1.4	3.7	2.2	0.3	0.3
**Primary education**	119	1.8	1.7	1.5	4.0	2.2	0.4	0.4
**Secondary education**	123	2.0	1.8	1.7	4.2	2.3	0.4	0.5
**Tertiary education**	116	2.1	1.7	1.7	4.0	2.2	0.4	0.5
**Women**	105	1.7	1.7	1.4	3.6	2.2	0.3	0.4
**Men**	136	2.5	1.9	2.0	4.6	2.4	0.5	0.5

* Results are based on only 15 respondents—thereby not statistically robust at a population-level.

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
