# Peer review of "Towards Win–Win Policies for Healthy and Sustainable Diets in Switzerland"

_nutrients, 2020, doi:10.3390/nu12092745_

Round 1

Reviewer 1 Report

This article provides an interesting analysis and relatively novel approach to assessing dietary nutrition and GHGE. It was overall well designed, uses a detailed and careful methodological approach to assessment, and is well written. In a few places there was a need for more explanation or clearer statement of key messages for reader. Below are more specific comments.

  • Intro/abstract
    • It would be helpful to more clearly and directly explain the study approach in the introduction/abstract.
  • Methods
    • I don’t understand the need for the “food prescriptions” work as described in the methods. Explain what is distinct to Switzerland or why there was a need for a sociology study to identify them, esp since most of the identified patterns were then excluded, leaving vegetarian, vegan and slimming, which are pretty standard dietary approaches for sustainability and health. (It is also not clear why gluten free is included, if it is not a predominant prescription, and not a top priority in most expert conceptualizations of health. That diet might be a marker for other cultural or health constructs not explored herein.)
    • LCI data – were system bounds consistently defined, if some studies included processing?
    • 137-45 “better meat” – this is an interesting argument regarding environmental impact variation by meat cut, although needs to be clarified with some succinct statement. I would say though, that if the goal is to capture what consumers mean when they say they eat ‘less but better’ meat, the term would refer to production method, such as grassfed, rather than the cut – which would make this argument inapplicable. (But it’s possible I'm misunderstanding the intent- needs clarity.)
    • This study uses a large number of acronyms which can make it difficult for readers to follow. The study has appeal for readers from two different fields who may not be familiar with each others’ acronyms, so I recommend reducing acronyms wherever possible.
    • I’m certainly familiar with DALYs but microdalys needs a little more explanation about how this differs from dalys – does it refer to different levels of disability, or just that 30 seconds is a small portion of time?
  • Results:
    • Would be helpful to move some basic descriptives from the appendix for context, e.g., start by summarizing # who adhered to each of the diet types, and extent to which self-described diets actually adhered to what might be expected criteria for this categorization. For example, to what extent were vegan diets free of animal products?
    • Given only 15 vegans, I recommend excluding. The fact that their results are exceptional on the first few analyses such as kcal, and they didn’t fully exclude animal products, suggests this may not be a meaningful or representative group of vegans. I think a separate study that has a larger sample is needed.
    • It is disappointing that sugar consumption apart from beverages wasn’t able to be included. But so be it!
    • In the nutritional assessment, I have questions about the summative nature of figure 3. Presuming it is designed to address the following concerns, please state that directly in methods, and ideally add a brief reminder in the framing text for the figure: Given that this study focuses on dietary patterns, to what extent can these dietary risk factors be understood independent of one another?  For example, can high consumption in one category compensate partly for low consumption in another (eg fruits vs vegetables). Also, in some cases, one food or dietary behavior may be contributing to multiple risk factors (milk, calcium is one example), so were the assessments underlying these risk factor categories designed to remove overlap? 
    • The nutritional assessment combines food groups and nutrients; consider separating these in different sections in the figures/tables.
    • Omega 3 fatty acids can come from sources besides seafood -is this meant to reflect only seafood omega 3’s?
    • Fig 4 is interesting, but I fear it could be used as a takeaway message from the study, leaving off the very different message that would be drawn based on a similar comparison of GHGs. Could you create a figure like this for ghg’s, or at least a figure that puts the two into conversation, such as based on table 2?
  • Discussion
    • What is the overarching novel contribution of this research? the findings on both health and nutrition are relatively common for their fields. So these would mostly not lead to practice change, although you do highlight an important gender issue for communication. To my mind, though there are other studies that bring together health and environmental data, this one does it in a relatively useful and sophisticated way. it would be useful to further draw that together, such as through the figure I suggest at the end, and through the framing of findings in the discussion and abstract. (If you see a different core contribution, then communicate that, but either way, there’s a need to emphasize the key takeaways more clearly/strongly.)
    • The first result mentioned in the discussion is the vegan men, and that’s too small of a subgroup to highlight.
    • (This statement from the conclusion is an important finding and could be a core message… “We found that consuming more whole grains has the potential to provide the highest health benefit 582 in Switzerland with little GHG consequence, and eating less processed meat offers the second highest 583 health benefit also with high benefits for GHG. Reductions in alcohol consumption are also a win-584 win for environmental and health outcomes.”)
    • Bring in this highly relevant study by Rose et al.. https://academic.oup.com/ajcn/article/109/3/526/5303906 This research by kanter et al. is also relevant in terms of self-defined sustainable diets. https://www.mdpi.com/2072-6643/12/2/489
    • the note about survey data not aligning with perceptions and not providing the info needed to evaluate health and sustainability is important.
    • The conclusion mentions “local diets,” and consumer perception about their sustainability benefits. It would be good to clarify that this is in fact not likely.

Author Response

Dear Reviewer 1,

Thank you for a very thorough and thoughtful review that has improved the quality of the manuscript. We have addressed your comments and made changes throughout. Best regards,

Alexi Ernstoff & Co-authors.

Review Report Form 1

Open Review

(x) I would not like to sign my review report

( ) I would like to sign my review report

English language and style

( ) Extensive editing of English language and style required

( ) Moderate English changes required

(x) English language and style are fine/minor spell check required

( ) I don't feel qualified to judge about the English language and style

            Yes       Can be improved           Must be improved         Not applicable

Does the introduction provide sufficient background and include all relevant references?

            ( )         (x)        ( )         ( )

Is the research design appropriate?

            (x)        ( )         ( )         ( )

Are the methods adequately described?

            (x)        ( )         ( )         ( )

Are the results clearly presented?

            (x)        ( )         ( )         ( )

Are the conclusions supported by the results?

            (x)        ( )         ( )         ( )

Comments and Suggestions for Authors

This article provides an interesting analysis and relatively novel approach to assessing dietary nutrition and GHGE. It was overall well designed, uses a detailed and careful methodological approach to assessment, and is well written. In a few places there was a need for more explanation or clearer statement of key messages for reader. Below are more specific comments.

Comment 1.1:    Intro/abstract

        It would be helpful to more clearly and directly explain the study approach in the introduction/abstract.

Answer and action: Thank you very much, we agree and we have rewritten the abstract to balance the approach with the results.

    Methods

Comment 1.2:         I don’t understand the need for the “food prescriptions” work as described in the methods. Explain what is distinct to Switzerland or why there was a need for a sociology study to identify them, esp since most of the identified patterns were then excluded, leaving vegetarian, vegan and slimming, which are pretty standard dietary approaches for sustainability and health. (It is also not clear why gluten free is included, if it is not a predominant prescription, and not a top priority in most expert conceptualizations of health. That diet might be a marker for other cultural or health constructs not explored herein.)

Answer: Thank you for this question. As this is an interdisciplinary project, we have the intention to bridge the gap between sociological work and environmental assessments, and agree the reason for this can be clarified better. We cannot ascertain whether these prescriptions are distinct to Switzerland as we have not studied them elsewhere.

Action: We have added lines to make these points clearer

In “Materials and Methods” section: “The purpose of including this step is to use cross-disciplinary knowledge to link social perceptions and practices to evidence if such practices are or are not healthy and sustainable. Furthermore, this step can help ascertain which data are (or are not) available to assess perceptions that may have implications for policy making.”

In “Discussion and limitations to the study” section: “In this study we first related sociological findings on predominate perceptions of healthy and sustainable diets (referred to as “prescriptions”) in Switzerland [6] to the dietary survey data available through the first Swiss National Survey, MenuCH. Investigation of the sociological context for the perception of health and sustainability combined with impact assessment is a developing area of inter-disciplinary work that can provide multiple layers of insight for policy and change making [31,32]. An initial finding from addressing the MenuCH data from a sociological perspective is that quantitative data are largely unavailable to assess the qualitative perceptions of healthy and sustainable diets. A clear example is the association of eating “balanced” or “locally” with health and sustainability; data were not available through menuCH (or otherwise) to assess the healthiness and sustainability of actual practices of people who aim to eat a “balanced” diet or “locally.” Other work suggests that Swiss in general (regardless of general knowledge of the Swiss food pyramid) are not consuming diets aligned with the food pyramid balance [10,13], and that “local” eating provides limited benefit for GHG reduction [33]. Another key findings from considering the sociological perspective is that vegetarian and vegan diets are perceived as healthier and more sustainable and were also found to have better GHG and health scores. Nevertheless, especially the health scores could be improved by food consumptions that are outside of these dietary prescriptions (i.e. increasing whole grains). Also the observed benefits of vegan and vegetarian diets were driven by key aspects of consumption i.e. consuming less processed meat) that could be practiced outside of this dietary prescription (e.g. in flexitarian or meat reduction diets, or simply diets that focus on eating at home as (data not shown) the majority of meat consumption was out-of-home. These finding suggests that perceptions that have high traction in day-to-day society with respect to health and sustainability may not be aligned with the evidence for specific win-win policies.”

 Comment 1.3:        LCI data – were system bounds consistently defined, if some studies included processing?

Answer and action: We acknowledge the “system boundaries” for different LCIs were not fully consistent due to both data availabilities and the inherent nature of food (some food is processed some is not). To more transparently address this as a limitation, we have added the following lines:

In the “Materials and Methods” section: “Due to the data available and the inherent differences between how foods are consumed in practice, the system boundaries of every life cycle inventory entry was not the same (i.e. some foods like “bread” include processing, where as other foods like “lettuce” do not), but represented the best available life cycle inventory data to match the item being consumed. Due to data unavailability we did not attempt to account for missing data across the life cycle of food items, and in this way the reported environmental impacts should be seen as an indication of the greenhouse gas emissions related to producing food to supply the swiss diet, and not the entire “food system” impacts which include e.g. transformation, transport and logistical (storage), and cooking impacts. The implications of this methodological choice on the result magnitude are further discussed in the discussion and limitations section of the study.

In “Discussion and limitations to the study” section:  “The overall magnitudes of GHG of all the diets are well-aligned with previous findings for average and special diets in other European countries [8]. As the purpose of this study was to compare across diets with a focus on impacts related to agricultural production, we did not include a detailed study of all life cycle processes such processing, packaging , refrigeration, preparation and washing (although some life cycle impacts were included for final products such as bottled water and bread). Without data available on the full value chain of food items, we assumed such additional impacts would be similar across diets from an environmental perspective e.g. vegans and the average Swiss would have similar practices related to food processing, packaging, refrigeration, cooking, and washing. If included these impacts would likely increase the footprint of the average diet by about 1.5 times [28,48],  adding about another 1 tCO2e/p/y. The differences across diets for life cycle processes such as food processing, refrigeration, cooking, and packaging related to various consumer practices could be interesting to investigate in a separate study.”

Comment 1.4:         137-45 “better meat” – this is an interesting argument regarding environmental impact variation by meat cut, although needs to be clarified with some succinct statement. I would say though, that if the goal is to capture what consumers mean when they say they eat ‘less but better’ meat, the term would refer to production method, such as grassfed, rather than the cut – which would make this argument inapplicable. (But it’s possible I'm misunderstanding the intent- needs clarity.)

Answer: “Better” meat in our previous qualitative study most often referred to “Swiss meat” and “organic”, but not to a precise means of production (beyond organic) such as grass fed. The notion of “better” assigned to meat cut is an assumption we make based on the high demand for more expensive cuts in Switzerland.

Action: We have elaborated a bit this argument and it reads as follows:

“The finding from sociological research that Swiss identify with eating “less and better meat” as “more healthy and sustainable” is difficult to quantify – for example consumers seemed to associate “better meat” as Swiss (not foreign) meat and potentially “organic”, yet at the same time there is demand for expensive (“better”) meat cuts in Switzerland. Given this consumer demand, one way to quantify “better meat” would be using the price of the meat cut (more expensive is higher quality and thus “better”); however this is contradictory to the sentiment that “better meat” is less impacting, as economic allocation and consequential thinking suggests that more expensive cuts can drive production. Furthermore, there are trade-offs and inconsistent evidence suggesting organic and pasture-based (“better”) beef and dairy production may have lower feed conversion efficiency and thus higher GHG emissions per kilogram of milk or meat; yet these systems have other benefits such as nutritional quality of the milk, animal welfare, and biodiversity [44–46]. This suggests that GHG is not the appropriate indicator to assess the benefits of “better meat” if this practice is describing consuming organic and/or pasture-based beef. Finally, although not possible to quantify in this study, there may be quantifiable GHG benefits of consuming “better meat” if this perception is leading to practices that avoid meat associated with deforestation (i.e. due to imported feed) and consuming a variety of cuts instead of more expensive ones (e.g. filet).”

Comment 1.5:         This study uses a large number of acronyms which can make it difficult for readers to follow. The study has appeal for readers from two different fields who may not be familiar with each others’ acronyms, so I recommend reducing acronyms wherever possible.

Answer and action: Thank you for this comment. We have removed the acronym “LCI” which refers to life cycle inventory, we have removed the acronym “DRF” dietary risk factor, and we have removed the acronym “GBD” which refers to global burden of disease. We have kept the acronym “GHG” for greenhouse gas which is commonly used in literature and beyond and DALY for disability adjusted life years which is also commonly used.

Comment 1.5:        I’m certainly familiar with DALYs but microdalys needs a little more explanation about how this differs from dalys – does it refer to different levels of disability, or just that 30 seconds is a small portion of time?

Answer and action: One µDALY is just 10-6 DALY - just like 1 microgram (µg),  is 10-6 grams. This unit is better suited to report the per-person nutritional effect of individual food items which is not on the level of losing years of life (e.g. a DALY). We have modified the text to read:

“The Swiss-specific DRFs are expressed in micro-Disability Adjusted Life Years (μDALY) per gram of food or nutrient and ultimately indicate the all-cause positive (avoided μDALY, e.g., negative DRF estimates) or detrimental health effects (positive DRFs). Just like a micrometer represents 1/1,000,000 of a meter, a microDALY represents 1/1,000,000 of a year, which results that a μDALY is 0.53 minutes of healthy life lost, or approximately 30 seconds.

    Results:

 Comment 1.6:        Would be helpful to move some basic descriptives from the appendix for context, e.g., start by summarizing # who adhered to each of the diet types, and extent to which self-described diets actually adhered to what might be expected criteria for this categorization. For example, to what extent were vegan diets free of animal products?

Answer: we have already discussed this as follows (no modification) – if you believe more should be added than this, or it should be moved please indicated in the next round of review.

“Our analysis revealed that self-identified diets may not be aligned in practice with what is generally recognized as the dietary prescription; for example, a subset of self-reported vegans (i.e. animal product-free diets) and vegetarians (i.e. meat-free diets) reported consuming dairy and meat respectively. Sixteen out of the 104 surveyed vegetarians reported eating meat leading to an average of 11 g/p/d. One self-identified vegan participant reported eating 56 g of processed meat, which led to a group daily average of 4 g/p/d, and 13 out of 15 vegan survey responses reported dairy consumption leading to an average of 40 g/p/d. It is unclear if these practices represent an issue or misunderstanding with the surveys or flexibility between self-declared dietary preferences and actual practices. Furthermore, other observed differences in vegan and vegetarian diets (e.g. more whole grains, less bottled water) suggest that the actual practices for these diets may be tied to lifestyle choices that are not restricted to only consumption of animal products.”

Comment 1.7:         Given only 15 vegans, I recommend excluding. The fact that their results are exceptional on the first few analyses such as kcal, and they didn’t fully exclude animal products, suggests this may not be a meaningful or representative group of vegans. I think a separate study that has a larger sample is needed.

Answer and action: We agree that these 15 surveys are likely not representative of vegans generally – however, the goal of the study was no necessarily to understand the total vegan population but analyze the MenuCH results. We find the results interesting to present alongside the other results. To avoid any over-interpretation we have added an Asterisk in Tables 1 & 2 to clarify. In-text results we no longer emphasize vegans but include “vegans and vegetarians” as one group.

Comment 1.8:         It is disappointing that sugar consumption apart from beverages wasn’t able to be included. But so be it!

Answer and action: Indeed data on impacts of sugar consumption is a limitation of the present state of knowledge and data available within the global burden of disease. There may be physiological reasons for this regarding satiation or matrix effects, e.g. see:  https://pubmed.ncbi.nlm.nih.gov/10878689/   & https://pubmed.ncbi.nlm.nih.gov/19248858/

We have added this sentence: “The global burden of disease work does not include risk factors where there is a lack of conclusive epidemiological evidence e.g. there is insufficient evidence to consider risks from general sugar consumption (apart from beverages) [35], as well as omega 3 sources outside of seafood.”

Comment 1.9:        In the nutritional assessment, I have questions about the summative nature of figure 3. Presuming it is designed to address the following concerns, please state that directly in methods, and ideally add a brief reminder in the framing text for the figure: Given that this study focuses on dietary patterns, to what extent can these dietary risk factors be understood independent of one another?  For example, can high consumption in one category compensate partly for low consumption in another (eg fruits vs vegetables). Also, in some cases, one food or dietary behavior may be contributing to multiple risk factors (milk, calcium is one example), so were the assessments underlying these risk factor categories designed to remove overlap?

        The nutritional assessment combines food groups and nutrients; consider separating these in different sections in the figures/tables.

Answer and action: Thank you for pointing this out. We have double checked the epidemiological results and ensured risks for double counting were removed when evidence was available. This resulted in one change in the results for calcium. Given we removed risks of double counting given the evidence in global burden of disease, we don’t find it useful to separate the foods and nutrients in the tables and figures. The following text was added to make the double counting issue more transparent: “When deriving the dietary risk factors care was taken to avoid double counting e.g. counting twice benefits or impacts that can occur for a nutrient and a food item that contains the nutrient. Due to the nature of epidemiological evidence being based on correlation, there remains a risk of double counting for nutrients such as saturated fat and products that contain saturated fat such as processed meat, although there is evidence that health impacts of processed meat are independent of saturated fat content [1]. Specifically, to avoid double counting we removed the benefits of calcium from milk (beverage) products which were accounted for separately in the “milk” category. Also to avoid double counting between fiber and fruits, vegetables, whole grains and legumes (f, v, w, l), we created a separate dietary risk factors for fiber f, v, w, l that does not include benefits of lowering heart disease risks which are already included in the benefits of consuming these whole plant foods.

Comment 1.10:        Omega 3 fatty acids can come from sources besides seafood -is this meant to reflect only seafood omega 3’s?

Answer and action: The factors from the GBD are specific to the " omega 3's from seafood" and the effect is therefore restricted to this food source. We have added a few sentences to clarify this “The vegan and vegetarian diets also lacked benefit (albeit a small difference) from omega 3 fatty acid from seafood (notably any benefits from omega 3 fatty acid from non-seafood sources were not included, as evidence is unavailable in the global burden of disease studies).” “The global burden of disease work does not include risk factors where there is a lack of conclusive epidemiological evidence e.g. there is insufficient evidence to consider risks from general sugar consumption (apart from beverages) [35], as well as omega 3 sources outside of seafood.”

Comment 1.11:        Fig 4 is interesting, but I fear it could be used as a takeaway message from the study, leaving off the very different message that would be drawn based on a similar comparison of GHGs. Could you create a figure like this for ghg’s, or at least a figure that puts the two into conversation, such as based on table 2?

Answer and action: This is a very important point, and we agree. We have created another figure to try to address this. Please see manuscript.

    Discussion

Comment 1.12:         What is the overarching novel contribution of this research? the findings on both health and nutrition are relatively common for their fields. So these would mostly not lead to practice change, although you do highlight an important gender issue for communication. To my mind, though there are other studies that bring together health and environmental data, this one does it in a relatively useful and sophisticated way. it would be useful to further draw that together, such as through the figure I suggest at the end, and through the framing of findings in the discussion and abstract. (If you see a different core contribution, then communicate that, but either way, there’s a need to emphasize the key takeaways more clearly/strongly.)

Answer and action: We have provided a paragraph to the discussion section to try to emphasize the novel findings.

“The novel contribution of this study is providing an analysis of self-reported practices in a given population to identify priority (and non-priority) policies that can simultaneously improve health and GHG emission. Similar work has been performed recently, for example for the United States [1]. A key novelty in our study is the separation of diets by reported gender and the use of the HENI [1] metric to derive health scores from global burden of disease. Unlike nutritional indices or qualitative indication of a food’s “healthiness”, HENI disease burden metrics help prioritize dietary shifts related to the relative order of magnitude of their benefit (or impact) on health for a given population. This study provided evidence that focusing on key food groups and men provides opportunities for priority win-win interventions. Mainly, this work identified the key win-win priorities as decreasing processed meat and alcohol consumption especially for men of higher education levels. Another identified priority is to increase whole grain consumption, e.g. to replace refined grains, across the population (including for people with diets low in animal products), yet this intervention would have less importance for reducing GHG emissions. This study also provided evidence that decreasing sugary sweetened beverages can provide win-win benefits for health and likely greenhouse gas reduction (i.e. if replaced with tap water), albeit is a much lower priority in comparison to other dietary shifts for Switzerland. Reducing red meat and dairy have key GHG benefits (depending on what may be substituted), yet according to the global burden of disease work have less importance for health in comparison to other components of the diet (see Figure 5). These findings suggest that policies towards dietary shifts to reduce red meat (especially for men of higher education) and reduce dairy for all individuals are priority for GHG emission that offer little trade-offs in the context of health – yet overall GHG emission and health impact would be dependent on what items are substituted. The impacts or benefits of what substitutes items that are decreased in the diet were not specifically assessed in this study, although other studies have shown benefits for plant-based substitutions (legumes, nuts and seeds, and whole grains) [1]. ”

Comment 1.13:        The first result mentioned in the discussion is the vegan men, and that’s too small of a subgroup to highlight.

Answer and action: We agree, and have changed this sentence to read “vegans and vegetarians.”

 Comment 1.14:       (This statement from the conclusion is an important finding and could be a core message… “We found that consuming more whole grains has the potential to provide the highest health benefit in Switzerland with little GHG consequence, and eating less processed meat offers the second highest health benefit also with high benefits for GHG. Reductions in alcohol consumption are also a win-win for environmental and health outcomes.”)

Answer and action: We agree, and have adapted the abstract and discussion to more emphasise this core messaging.

Comment 1:15:        Bring in this highly relevant study by Rose et al.. https://academic.oup.com/ajcn/article/109/3/526/5303906 This research by kanter et al. is also relevant in terms of self-defined sustainable diets. https://www.mdpi.com/2072-6643/12/2/489

        the note about survey data not aligning with perceptions and not providing the info needed to evaluate health and sustainability is important.

        The conclusion mentions “local diets,” and consumer perception about their sustainability benefits. It would be good to clarify that this is in fact not likely.

Answer/Action: We have included these citation and more discussion about the “local” issue, as seen in these lines added:

“The novel contribution of this research is providing an analysis of self-reported practices in a given population to identify priority (and non-priority) policies that can improve health and GHG emission. Similar work has been performed recently, for example for the United States [1].”

 “Investigation of the sociological context for the perception of health and sustainability combined with impact assessment is a developing area of inter-disciplinary work that can provide multiple layers of insight for policy and change making [1,2].”

 “A clear example is the association of eating “balanced” or “locally” with health and sustainability; data were not available through menuCH (or otherwise) to assess the healthiness and sustainability of actual practices of people who aim to eat a “balanced” diet or “locally.” Other work suggests that Swiss in general are not consuming balanced diets [1,2], and that “local” eating provides limited benefit for GHG reduction [1].”

Reviewer 2 Report

I have reviewed the article manuscript titled “Screening the healthiness and greenhouse gas emissions of reported Swiss diets to inform targeted policies” submitted for publication in the Nutrients journal.

The paper presents a combined analysis of health and environmental effects of various Swiss diets. The methodology is described in detail, including its limitations. The research produced several interesting results, some of them in line with previous research, some of them interestingly new or contradicting to previous analyses. The paper thus makes a valuable contribution.

I have mainly comments regarding the presentation of the paper, which I consider minor revisions. I will detail my suggestions below in the order I encountered them reading the paper (i.e. not in order of importance):

  • I find the title currently a bit bland. It refers mainly to the methodology, the way the study was carried out (“screening”). I suggest the title be revised to refer also to some of the main results of the study, for instance “Screening of healthiness and greenhouse gas emissions of Swiss diets suggests targeting policies on men”
  • Abstract: it is mentioned that “Results demonstrated Swiss diets ranged between 1.1-2.6 tonnes of 21 CO2e/person/year.” The figure as such does not necessarily say much to all readers, so this could be contextualized by comparing it to the targets by Swiss government, or making an international comparison.
  • The study is, in part, motivated by sustainability concerns of diets and food production. Then, GHG emissions are used as an indicator for the environmental impact. Sustainability of food cannot be reduced one-to-one to greenhouse gas emissions. In the introduction a couple of sentences could be added (around rows 61-62) explaining why GHG emissions were chosen to be used, and the limitations of this choice (e.g. eutrophication and resulting biodiversity decline are also important environmental impacts of food production).
  • At a few instances there are some literature references in the author+date –format, even though the journal uses numbered endnote format (an example on row 80, 168). Please revise.
  • The word “survey” is used in the paper in two different meanings: to refer to the whole survey study, and to individual survey responses. An individual response does not constitute a survey, so please revise the use of the term. e.g. row 88: “A total of 3,860 different surveys were completed.” should read something like “A total of 3,860 individual survey responses were completed/acquired.” Also lines 236-237: surveys -> survey responses. Line 259.
  • An unclear sentence on rows 102-104: “To note MenuCH provides the information someone has self-identified as following a special diet, but practices may deviate from the widely accept definition of the diet”, also, the described phenomenon is commonly known as “reporting bias”.
  • Row 109: “we also assessed the average diet as a reference”, it was not clear to me what is this average diet, where does it come from? Please revise. (Also earlier in the paper the authors criticize studies addressing average diets, whereas the study reported here addresses actual practices – slightly contradicting claims, if average diet is taken as a measure after all?)
  • a typo on row 124: good -> food
  • 137: rational -> rationale
  • 174: “Juices and any foods where we the” -> delete “we”
  • the paper uses italics in the text somewhat randomly, for instance on line 190, 306, 439. I suggest all italics to be removed, except the subtitles and research questions.
  • the text contains some internal references that are broken and are replaced with text “Error! Reference source not 262 (lines 262, 302, 332, 351) Please revise.
  • typo row 344: consumer -> consume/consumed
  • At the end of the first paragraph the discussion refers to emotions and moralities of meat consumption, but the thought seems to be left in the air. What do the authors want to say with this? (lines 432-436)
  • lines 441-444: “This suggests that the reason special diets were more beneficial to health than the average diet was because of other lifestyle (consumption) choices, beyond the restriction of meat or animal product consumption.” Here only food consumption was studied, so the paper cannot say anything on other lifestyle choices. Some rewording?
  • 447: “Given the emotional discourse surrounding vegan and vegetarian diets in Switzerland”. the international reader does not know the discourse concerning veganism in Switzerland, so this has to be explained, or left out. As such, it gives an impression the authors are belittling the discourse for being “emotional”.
  • Some of the results of the study are very interesting as they contradict previous research, such as reducing meat per se is perhaps not the best way to reduce environmental & health impacts of food, but to increase the use of wholegrain foods. It could be specified though, whether wholegrains should be added to current diets, or whether they should replace something (e.g. rows 581-583).
  • And important, and somewhat startling, result of the study is that even if all the Swiss would become vegans, the environmental impact of diet would still be beyond sustainable. This is a good indication of the magnitude of food transformation necessary, on systemic level (including e.g. food waste as discussed by authors).
  • the authors seem surprised that (male) vegans consume more protein, mass and energy than those following “average” diet. One reason for the health score for vegan (and other “special”) diets is likely that as one starts to follow a diet, it is necessary to become more informed than average on the nutrition contents of different foods.
  • An important suggestion of the study is to target dietary interventions according to gender. While the gender differences in health and environmental performance are quite well known, it is not common to suggest targeting policy to men, as they have higher consumption-related environmental impact.
  • lines 369-370: “Processed meat consumption by the average man was associated with an increased risk of 24 μDALY/p/d which can be interpreted as approximately 13 minutes of healthy life lost each day.” I found this information rather striking, and it attests to the efficiency of the μDALY/p/d as a potential tool in dietary advice. While scaring people of dying prematurely might not be the most ethical or even efficient dietary policy, maybe a short discussion could be included here as to the efficiency of this information for consumer choices.

Thanks for the opportunity of revising your interesting paper.

Author Response

Dear Reviewer 2,

Thank you for a very thorough and thoughtful review that has improved the quality of the manuscript. We have addressed your comments and made changes throughout. Best regards,

Alexi Ernstoff & Co-authors

Reviewer 2

Open Review

(x) I would not like to sign my review report

( ) I would like to sign my review report

English language and style

( ) Extensive editing of English language and style required

( ) Moderate English changes required

(x) English language and style are fine/minor spell check required

( ) I don't feel qualified to judge about the English language and style

            Yes       Can be improved           Must be improved         Not applicable

Does the introduction provide sufficient background and include all relevant references?

            ( )         (x)        ( )         ( )

Is the research design appropriate?

            (x)        ( )         ( )         ( )

Are the methods adequately described?

            (x)        ( )         ( )         ( )

Are the results clearly presented?

            (x)        ( )         ( )         ( )

Are the conclusions supported by the results?

            ( )         (x)        ( )         ( )

Comments and Suggestions for Authors

I have reviewed the article manuscript titled “Screening the healthiness and greenhouse gas emissions of reported Swiss diets to inform targeted policies” submitted for publication in the Nutrients journal.

The paper presents a combined analysis of health and environmental effects of various Swiss diets. The methodology is described in detail, including its limitations. The research produced several interesting results, some of them in line with previous research, some of them interestingly new or contradicting to previous analyses. The paper thus makes a valuable contribution.

I have mainly comments regarding the presentation of the paper, which I consider minor revisions. I will detail my suggestions below in the order I encountered them reading the paper (i.e. not in order of importance):

Comments 2.1:    I find the title currently a bit bland. It refers mainly to the methodology, the way the study was carried out (“screening”). I suggest the title be revised to refer also to some of the main results of the study, for instance “Screening of healthiness and greenhouse gas emissions of Swiss diets suggests targeting policies on men”

Answer and action: Thank you for this suggestion. We suggest a new title of: “Towards win-win policies for healthy and sustainable diets in Switzerland”

 Comments 2.2:   Abstract: it is mentioned that “Results demonstrated Swiss diets ranged between 1.1-2.6 tonnes of 21 CO2e/person/year.” The figure as such does not necessarily say much to all readers, so this could be contextualized by comparing it to the targets by Swiss government, or making an international comparison.

Answer and action: Thanks you, we agree and have changed the abstract to read: “Swiss diets, including vegetarian, ranged between 1.1-2.6 tonnes of CO2e/person/year, above the Swiss federal recommendation 0.6 tonne CO2/person/year for all consumption categories. This suggests that only changing food consumption practices will not suffice towards achieving carbon reduction targets: systemic changes to food provisioning processes are also necessary.”

Comments 2.3:    The study is, in part, motivated by sustainability concerns of diets and food production. Then, GHG emissions are used as an indicator for the environmental impact. Sustainability of food cannot be reduced one-to-one to greenhouse gas emissions. In the introduction a couple of sentences could be added (around rows 61-62) explaining why GHG emissions were chosen to be used, and the limitations of this choice (e.g. eutrophication and resulting biodiversity decline are also important environmental impacts of food production).

Answer and action: We agree, and have added these lines: “Because this study focuses on different diets documented by MenuCH (which does not report the location of food production but only food product type), GHG was chosen as a robust and comparable sustainability indicator because it is less sensitive to location of food production unlike biodiversity and water impacts. Furthermore, GHG is relevant for current Swiss policy decisions regarding GHG targets.”

Comments 2.4:    At a few instances there are some literature references in the author+date –format, even though the journal uses numbered endnote format (an example on row 80, 168). Please revise.

Answer and action: We have kept author + date with the numbered endnote in the text only when we are referring specifically a to study – we feel this is appropriate. We added the end note citation directly after the quote author to be more clear.

Comments 2.5:    The word “survey” is used in the paper in two different meanings: to refer to the whole survey study, and to individual survey responses. An individual response does not constitute a survey, so please revise the use of the term. e.g. row 88: “A total of 3,860 different surveys were completed.” should read something like “A total of 3,860 individual survey responses were completed/acquired.” Also lines 236-237: surveys -> survey responses. Line 259.

Answer and action: Thank you, we have addressed this issue.

 Comments 2.6:   An unclear sentence on rows 102-104: “To note MenuCH provides the information someone has self-identified as following a special diet, but practices may deviate from the widely accept definition of the diet”, also, the described phenomenon is commonly known as “reporting bias”.

Answer and action: Thank you, we have adapted the sentence to clarify this point, reading as: “Note that MenuCH data had inconsistencies regarding declarations of special diets (e.g., vegan), and the actual foods reported ( e.g. vegans declared animal product consumption).”

Comments 2.7:    Row 109: “we also assessed the average diet as a reference”, it was not clear to me what is this average diet, where does it come from? Please revise. (Also earlier in the paper the authors criticize studies addressing average diets, whereas the study reported here addresses actual practices – slightly contradicting claims, if average diet is taken as a measure after all?)

Answer and action: We have removed this line as it is confusing and adds no value.

Comments 2.8s:

    a typo on row 124: good -> food

    137: rational -> rationale

    174: “Juices and any foods where we the” -> delete “we”

Answer and action: corrected the text according to comments

Comments 2.9:    the paper uses italics in the text somewhat randomly, for instance on line 190, 306, 439. I suggest all italics to be removed, except the subtitles and research questions.

Answer and action: Italics were removed except in subtitles and research questions.

Comments 2.10:    the text contains some internal references that are broken and are replaced with text “Error! Reference source not 262 ” (lines 262, 302, 332, 351) Please revise.

    typo row 344: consumer -> consume/consumed

Answer and action: Thank you, this has been corrected.

Comments 2.11:    At the end of the first paragraph the discussion refers to emotions and moralities of meat consumption, but the thought seems to be left in the air. What do the authors want to say with this? (lines 432-436)

Answer and action: We have elaborated this thought to read as :  “The priority levers of change identified in this study (e.g. decreasing processed meat and alcohol for men and increasing whole grains for all) are not key aspects of the current discourse at various stakeholder and scientific events attended by the authors or the studied “pro” or “no” meat public discourse in Switzerland which tends towards specific moral and emotional messaging. Generally discourse around food tends to focus on processed foods, and “pro” or “no” meat consumption in general. Emotional messaging and debate regarding meat (and animal product) consumption in Switzerland tends to use associations of national pride to promote meat and cheese, or use imagery leading to disgust to denounce animal husbandry [1]. Further study could consider how such emotional messaging may (or may not) change practice in comparison to other interventions such as food availability in relation to daily mobility and in schools, and beyond. Investigation of emotional discourse could also shed light if the communication of health impacts (such as those found in this study) and emotions that may elicit is or is not helpful in changing dietary choices.”

Comments 2.12:    lines 441-444: “This suggests that the reason special diets were more beneficial to health than the average diet was because of other lifestyle (consumption) choices, beyond the restriction of meat or animal product consumption.” Here only food consumption was studied, so the paper cannot say anything on other lifestyle choices. Some rewording?

Answer and action: Thank you for spotting this, we have reworded this section to read as “For our second research question on most influential factors, disease burden scores for vegetarians, vegans, gluten-free and other special diets were largely influenced by whole grains, nuts and seeds, processed meat and alcohol. This finding, along with findings regarding differential use of bottled water between special diets (i.e. vegetarians consumed less bottled water), suggests that there may be additional consumption choices important for health and/or sustainability performed by people self-declaring special diets, but that are beyond the definition of the special diet. Future study could be expanded to understand what drives this larger set of practices or lifestyle choices for example to also consider people’s mobility or social interactions.”

 Comments 2.13:   447: “Given the emotional discourse surrounding vegan and vegetarian diets in Switzerland”. the international reader does not know the discourse concerning veganism in Switzerland, so this has to be explained, or left out. As such, it gives an impression the authors are belittling the discourse for being “emotional”.

Answer and action: We agree and have changed this section as: “The priority levers of change identified in this study (e.g. decreasing processed meat and alcohol for men and increasing whole grains for all) are not key aspects of the current discourse at various stakeholder and scientific events attended by the authors or the studied “pro” or “no” meat public discourse in Switzerland which tends towards specific moral and emotional messaging. Generally discourse around food tends to focus on processed foods, and “pro” or “no” meat consumption in general. Emotional messaging and debate regarding meat (and animal product) consumption in Switzerland tends to use associations of national pride to promote meat and cheese, or use imagery leading to disgust to denounce animal husbandry [1]. Further study could consider how such emotional messaging may (or may not) change practice in comparison to other interventions such as food availability in relation to daily mobility and in schools, and beyond. Investigation of emotional discourse could also shed light if the communication of health impacts (such as those found in this study) and emotions that may elicit is or is not helpful in changing dietary choices.”

Comments 2.14:    Some of the results of the study are very interesting as they contradict previous research, such as reducing meat per se is perhaps not the best way to reduce environmental & health impacts of food, but to increase the use of wholegrain foods. It could be specified though, whether wholegrains should be added to current diets, or whether they should replace something (e.g. rows 581-583).

Answer and action: The evidence from this study pin-points “win-win” priority situations. Although reducing red meat is a “win-win” it is much less than reducing processed meat (from a health perspective red meat it is nearly negligible where processed meat is a leading driver of risk). We don’t find that this finding is a contradiction of past work but is a way to prioritize interventions. We have restructured large portions of the discussion to address this point.

Comments 2.15:    And important, and somewhat startling, result of the study is that even if all the Swiss would become vegans, the environmental impact of diet would still be beyond sustainable. This is a good indication of the magnitude of food transformation necessary, on systemic level (including e.g. food waste as discussed by authors).

Answer and action: Thank you very much for this point, which we have further emphasized in the conclusion and also added to the abstract.

Comments 2.16:    the authors seem surprised that (male) vegans consume more protein, mass and energy than those following “average” diet. One reason for the health score for vegan (and other “special”) diets is likely that as one starts to follow a diet, it is necessary to become more informed than average on the nutrition contents of different foods.

Answer and action: We have removed the subjective word “surprised” and agree there may be many reasons for this outside the scope of this study as discussed – no others changes have been made.

Comments 2.17:    An important suggestion of the study is to target dietary interventions according to gender. While the gender differences in health and environmental performance are quite well known, it is not common to suggest targeting policy to men, as they have higher consumption-related environmental impact.

Answer and action: Thank you very much for this comment, we further emphasize this point throughout the manuscript (abstract, discussion, and conclusion).

Comments 2.18:    lines 369-370: “Processed meat consumption by the average man was associated with an increased risk of 24 μDALY/p/d which can be interpreted as approximately 13 minutes of healthy life lost each day.” I found this information rather striking, and it attests to the efficiency of the μDALY/p/d as a potential tool in dietary advice. While scaring people of dying prematurely might not be the most ethical or even efficient dietary policy, maybe a short discussion could be included here as to the efficiency of this information for consumer choices.

Answer and action: We agree. We have reformulated the discussion to include a line that reads as:

“Further study could consider how such emotional messaging may (or may not) change practice in comparison to other interventions such as food availability in relation to daily mobility and in schools. Investigation of emotional discourse could also shed light if the communication of health impacts (such as those found in this study) is or is not helpful in changing dietary choices.”

Thanks for the opportunity of revising your interesting paper.

Answer: Thank you for your thoughtful and helpful review comments that have improved the quality of the manuscript.
